# FEYNMAN-KAC OPERATOR EXPECTATION ESTIMATOR

## ABSTRACT

The Feynman-Kac Operator Expectation Estimator (FKEE) is an innovative method for estimating the target Mathematical Expectation $\mathbb{E}_{X \sim P}[f(X)]$ without relying on a large number of samples, in contrast to the commonly used Markov Chain Monte Carlo (MCMC) Expectation Estimator. FKEE comprises diffusion bridge models and approximation of the Feynman-Kac operator. The key idea is to use the solution to the Feynmann-Kac equation at the initial time $u(x_0, 0) = \mathbb{E}[f(X_T)|X_0 = x_0]$. We use Physically Informed Neural Networks (PINN) to approximate the Feynman-Kac operator, which enables the incorporation of diffusion bridge models into the expectation estimator and significantly improves the efficiency of using data while substantially reducing the variance. Diffusion Bridge Model is a more general MCMC method. In order to incorporate extensive MCMC algorithms, we propose a new diffusion bridge model based on the Minimum Wasserstein distance. This diffusion bridge model is universal and reduces the training time of the PINN. FKEE also reduces the adverse impact of the curse of dimensionality and weakens the assumptions on the distribution of $X$ and performance function $f$ in the general MCMC expectation estimator. The theoretical properties of this universal diffusion bridge model are also shown. Finally, we demonstrate the advantages and potential applications of this method through various concrete experiments, including the challenging task of approximating the partition function in the random graph model such as the Ising model.

## 1 INTRODUCTION

### 1.1 BACKGROUND

Markov Chain Monte Carlo (MCMC) is a prevalent computational method used in fields such as statistics, machine learning, and computational science. It is primarily applied for sampling from complex distributions, Bayesian inference, and optimization (Hesterberg (2002); Ahmed (2008)). MCMC algorithms are typically divided into two categories: those that sample from the target distribution and those that estimate the statistical characteristics of the target distribution, such as the expectation. While alternative methods to traditional MCMC samplers, like generative models (Song & Ermon (2019)) and diffusion models (Ho et al. (2019)), have been explored, MCMC remains the standard for expectation estimation. However, traditional MCMC estimators, based on the law of large numbers (**LLN**) and the ergodic theorem of Markov chains (**ETMC**), face limitations in data efficiency and **impose complex constraints on the distribution $P$ and performance function $f$**. Thus, developing algorithms that overcome these limitations by integrating modern sampling methods with deep learning is of great importance.

### 1.2 MOTIVATION

**Advantages of MCMC algorithm:** MCMC algorithms are effective for sampling from target distributions and are accompanied by two types of expectation estimators. The first type, based on LLN, uses averages from multiple samples at the terminal time of the Markov chain. The second type, based on ETMC, averages values along the path of the Markov chain. These methods utilize statistical principles effectively, particularly for high-dimensional distributions, mitigating the curse of dimensionality in integral approximations.

**Disadvantages of MCMC algorithm**: Despite their advantages, MCMC algorithms are not optimal for expectation estimation. The efficiency of MCMC estimators depends on the distribution $P$ and the function $f$. Different MCMC algorithms are required for different $P$, and due to burn-in periods, MCMC algorithms often waste many points. Additionally, the sample size $N$ shoule be large enough to achieve accurate estimates, leading to variances on the order of $\mathcal{O}(\sqrt{N})$. Quasi-Monte Carlo methods offer variances on the order of $\mathcal{O}(N^{\frac{1}{2}+\delta})$ with $\delta \leq \frac{1}{2}$ (Caflisch (1998)), but this diminishes efficiency and introduces bias. Error probabilities can be estimated through concentration inequalities (Lugosi (2003)), but these depend on the Lipschitzian norm of $f$.

Estimating mathematical expectations is crucial in both machine learning and statistics. A unified expectation estimator is theoretically and practically significant. Therefore, we focus on two essential questions:

*(A) Is it possible to unify most existing MCMC algorithms into a cohesive framework to create a universal sampler for expectation estimation?*

*(B) How can we develop a universal expectation estimator that leverages samples from universal samplers for accurate expectation estimates without relying on post-processing or specialized methods?*

**For the first question**: We propose the following solutions. we know the Markov model is determined by the transfer density of one step. The transition density function associated with the discrete Markov chain generated by the MCMC algorithm can be interpreted as the transition density function of a specific stochastic differential equation (SDE) of Markov properties. In this study, we refer to this SDE as the **diffusion bridge model**. This encompasses a broad class of SDEs that share identical transition densities with the Markov chains in the MCMC algorithm. The distribution of the terminals in such SDEs aligns with a predefined target distribution, which can take the form of discrete points or a probability density function. Moreover, the starting point of this SDE can be either arbitrary or fixed.

**For the second question**: In the context of the diffusion bridge model, we can view the expectation estimation problem as a decoding problem. It is easy to observe that MCMC is not the optimal decoding method because a substantial number of burn-in samples go to waste when estimating mathematical expectations by using the MCMC algorithm. However, these samples harbor valuable information, specifically pertaining to the gradient information of the drift and diffusion coefficients along the paths derived from the SDE. We capitalize on this information by integrating it through the Physics-Informed Neural Network (PINN) approach (Sharma & Shankar (2022); Raissi et al. (2019); Yuan et al. (2022)). This process, akin to approximating the Feynman-Kac Operator, is referred to as solving the **Feynman-Kac model**. Notably, this approximation is meshless and effectively overcomes the curse of dimensionality. By

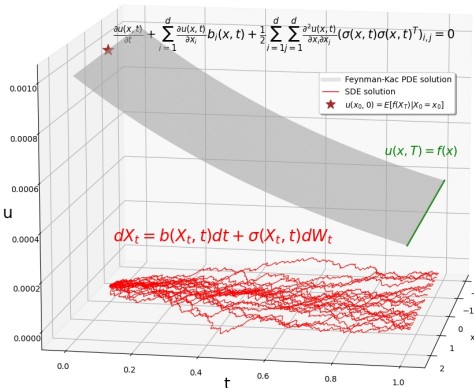

Figure 1: The expectation is obtained by simulating the SDE and then solving the PDE

amalgamating different combinations of the aforementioned models, we derive the **Feynman-Kac Operator Expectation Estimator** (FKEE) in Figure 1. In the framework of FKEE, the diffusion bridge model can extend the broader category of distributions $P$, while the Feynman-Kac model can extend the broader category of performance functions $f$.

**Our contributions can be summarized as follows:**

- **Expanding the Scope of Expectation Estimators:** Our approach enhances the efficiency of Markov chains, requiring fewer assumptions and not relying on LLN or the Markov ergodic theorem.

- **Introducing a more versatile diffusion bridge model:** We introduce a highly adaptable diffusion bridge model. This model not only allows for the specification of target distribu-

tions at terminal moments but also facilitates the reconstruction of the entire Markov chain. It can be employed in conjunction with the Feynman-Kac model for expectation estimation, as well as independently for resampling target distributions to estimate expectations.

- **Impacts on various fields combined with diffusion model:** We offer an alternative interpretation of diffusion models and mathematical expectation computations, where the diffusion model functions as the encoder and FKEE serves as the decoder. Existing diffusion models provide a powerful paradigm for learning data distributions, broadening the category of distributions $P$, while FKEE broadens the category of performance functions $f$. This enhances the utility of a broad class of existing diffusion models (based on SDE samplers), leading to various interesting applications in fields such as statistics and machine learning.

## 2  MAIN METHODOLOGIES

### 2.1  NOTATIONS

Let $\mu_t$ represent the distribution of $X_t$ and $\hat{\mu}_t = \frac{1}{N} \sum_{i=1}^{N} \delta_{X_t^i}$ denote the empirical distribution of $X_t^i, 1 \leq i \leq N$, where $\delta_X$ is the Dirac measure at $X$. $A_{i,j}$ denotes the elements of row $i$ and column $j$ of the matrix $A$, $diag(A)$ represents the diagonal matrix of matrix $A$ and $diag(A)_i$ denotes the i-th element on the main diagonal of the diagonal matrix of $A$. $C^2$ denotes the space of continuous functions with second order derivatives. Denote $Y \sim P := \mu^*$ by the target distribution.

### 2.2  DIFFUSION BRIDGE MODEL

The sampling methods mentioned in related work can be generalized into a common framework: for most MCMC sampling methods, we can consider using a Markov-type SDE as follows:

$$dX_t = b(X_t, t)dt + \sigma(X_t, t)dW_t, \qquad X_0 = x_0, \tag{1}$$

where $b : \mathbb{R}^d \times [0, T] \to \mathbb{R}^d$ is a vector-valued function , $\sigma : \mathbb{R}^d \times [0, T] \to \mathbb{R}^{d \times d}$ is a matrix-valued function, and $\{W_t\}_{t \geq 0}$ is a Brownian motion taking values on $\mathbb{R}^d$.

For a non-stationary diffusion model, these coefficients $b$ and $\sigma$ must satisfy certain regularity conditions to ensure the existence and uniqueness of a strong solution. For diffusion models with a stationary distribution, the uniqueness of the stationary distribution must hold. For a given distribution $P$ or a set of discrete points from $P$, we encode the information of $P$ into $(X_0, b, \sigma, T)$. This encoding algorithm should ensure that the distance between $\mu_T$ and $P$ is sufficiently small. The encoding loss to be minimized is:

$$\mathcal{L}_e = \mathcal{W}_2(\mu_T, \mu^*), \tag{2}$$

where $\mu_T$ is the measure of solution of SDE (1) and $\mathcal{W}_2$ is *2-Wasserstein distance*. The encoding loss $\mathcal{L}_e$ consists of two main components: structural loss and discretization loss. Structural loss is typically induced by the accuracy of $(b, \sigma)$, and discretization loss is usually due to the need for a sufficiently large $T$ and the numerical discretization of the SDE. According to the triangle inequality, the error can be decomposed as:

$$\mathcal{L}_e \leq \underbrace{\mathcal{W}_2(\mu_T, \bar{\mu}_T)}_{\text{discretization loss}} + \underbrace{\mathcal{W}_2(\bar{\mu}_T, \mu^*)}_{\text{structural loss}}, \tag{3}$$

where $\bar{\mu}_T$ is the distribution of numerical solution. The encoding loss depends on whether the distribution $P$ has an explicit density.

Specifically, if the density of $P$ is known, the structural loss has two components: the approximation error of $(X_0, b, \sigma)$ itself and the asymptotic error. The asymptotic error exists only in certain methods when the target distribution is the stationary distribution, such as in Langevin MCMC, where the error of $(X_0, b, \sigma)$ is zero and the asymptotic error decreases exponentially. However, discretization loss arises from the SDE solver.

If the density of $P$ is unknown, the constructed SDE will exhibit both types of losses. The core focus of the diffusion model is to minimize these two losses. The error in $(b, \sigma)$ is controlled by a specific loss function, while discretization loss is controlled by minimizing $T$ as much as possible and using a high-precision SDE solver.

In this paper we propose a diffusion bridge model that minimizes the encoding loss through the use of parameterized tuples $(X_0, b, \sigma)$. This method is similar to the Neural SDE (Tzen & Raginsky (2019); Kidger et al. (2021)). Specifically, we use the following Neural SDE:

$$dX_t = b_{\theta_1}(X_t, t)dt + \sigma_{\theta_2}(X_t, t)dW_t, \qquad X_{0,\theta_3} = x_{0,\theta_3}, \qquad (4)$$

where $\mathcal{P}_\theta = (X_{0,\theta_3}, b_{\theta_1}, \sigma_{\theta_2})$ is a neural network, typically a multi-layer perceptron (MLP) with the *tanh* activation function. Here, the time $T$ and time step $h$ are given in advance for the SDE solver. Diffusion bridge model matching means that we use neural network methods to find the appropriate $(X_0^*, b^*, \sigma^*)$ such that the distribution of $X_T$ at the moment $T$ is just the given target distribution $P$. We need to categorise the target distribution to determine the matching method. This depends on whether the target distribution has an explicit probability density function.

**Encoding loss for diffusion bridge models**

We examine the error of the diffusion bridge model. Unlike other design loss functions, we aim to control both errors in loss (3) simultaneously. Different loss functions correspond to different problems, necessitating the classification of the target distribution.

**Only a few discrete observations:** We propose a matching algorithm that deals with only a subset of discrete points from the target distribution $P$. Specifically, we employ a diffusion bridge model to parameterize $(X_{0,\theta}, b_\theta, \sigma_\theta)$ using a neural network. Given the empirical distribution of the target $\hat{\mu}^*$, we simulate $N$ trajectories of Brownian motion and use the Euler-Maruyama method (Platen (1999)) to obtain the solution $\bar{X}_T$. Subsequently, we match the obtained solutions to the given points and utilize the Wasserstein distance loss function:

$$\mathcal{P}_\theta{}^* = \arg\min_{\mathcal{P}_\theta \in \Theta} \mathcal{W}_2(\hat{\bar{\mu}}_T, \hat{\mu}), \qquad (5)$$

where $\hat{\bar{\mu}}_T$ (and $\hat{\mu}$ respectively) is the empirical distribution of independent identical copies of $\bar{X}_T$ (and $\mu^*$ respectively). Given $T$ and $h$, we can estimate the discretization loss and control the structural loss through the Wasserstein distance loss. This method has two additional applications:

*(i) Resample (Generate) samples:* For a set of high-quality samples (not within the burn-in period of MCMC), this method can be used for resampling. By matching a diffusion bridge model to the given points, we can simulate the SDE to obtain more samples. High-quality samples can also be obtained through other methods, such as Perfect Sampling (Djurić et al. (2002)).

*(ii) Matching Markov chains and generating more Markov chains quickly:* For trajectories $Y_i^N$ of $N$ independently run Markov chains obtained from MCMC algorithms $Y_i \sim \mu_{t_i}^*$ where $i \leq M$, we aim to find a set of $(X_0, b, \sigma)$ such that $\bar{X}_{t_i}$ and $Y_i$ are close at $M$ moments in the sense of the Wasserstein distance. Here, $\bar{X}_{t_i}$ is the solution to the SDE defined by $(X_0, b, \sigma)$. This can be achieved by optimizing the Wasserstein distance loss:

$$\mathcal{P}_\theta{}^* = \arg\min_{\mathcal{P}_\theta \in \Theta} \sum_{i=1}^{M} \mathcal{W}_2(\hat{\bar{\mu}}_{t_i}, \hat{\mu}_{t_i}^*), \qquad (6)$$

where $\hat{\bar{\mu}}_{t_i}$ and $\hat{\mu}_{t_i}^*$ are empirical distributions corresponding to $\bar{X}_{t_i}$ and $Y_i$ respectively. This process allows for matching either a segment or the entire Markov chain, potentially starting the matching process from a later moment to minimize reliance on points within the burn-in period.

---

**Algorithm 1** Diffusion bridge model (DBM)

---

**Input**: Initial value:$X_{0,\theta_3}$, Brownian motion:$W_t$. Neural network:$b_{\theta_1}(x,t)$, $\sigma_{\theta_2}(x,t)$. $\varepsilon$ is the required error threshold. The given data point $\{Y_T^k\}_{k=1}^N$ follows the distribution of $Y_T$.
**Output**: $X_t, b(t, X_t), \sigma(t, X_t)$.
  1: Simulate $X_t$ by Euler-Maruyama method.
  2: Calculate loss $\mathcal{L}$ in (5).
  3: **if** Match the whole Markov chain **then**
  4:     Calculating the loss $\mathcal{L}$ in (6). {The data points $\{Y_i^k\}_{k=1}^N$ are from Markov chains}
  5: **end if**
  6: Update parameters $\theta_1, \theta_2, \theta_3$.
  7: **if** $\mathcal{L} < \varepsilon$ **then**
  8:     End of training.
  9: **end if**

---

This algorithm is a simplified version of a more detailed one available in the Appendix 10.

In the following we present some theoretical results with the proofs given in the Appendix. We first provide an estimate for the discrete loss.

**Theoretical results**

**Theorem 2.1.** *Assuming that $b$ and $\sigma$ are L-lipschitz functions and Linear growth, more precise in (8.1). SDE solver is the Euler-Maruyama method ,we can obtain the following estimate:*

$$\mathcal{W}_2(\mu_T, \bar{\mu}_T) \leq Ch^{\frac{1}{2}} \exp\left(4L^2 T\right), \tag{7}$$

*where $C$ depends on $X_0$, but it is independent of $h$. We can pre-select suitable $T$ and $h$ to control this error.*

*Proof.* In the Appendix 8.3. $\qquad\square$

For convenience we use the following notations to indicate that the measures depend on the parameter. $\mu_T := \mu_T^{\mathcal{P}_\theta}$, $\bar{\mu}_T := \bar{\mu}_T^{\mathcal{P}_\theta}$, $\hat{\mu}_T := \hat{\mu}_T^{N,\mathcal{P}_\theta}$, $\hat{\bar{\mu}}_T := \hat{\bar{\mu}}_T^{N,\mathcal{P}_\theta}$, $\hat{\mu}_{t_i} := \hat{\mu}_{t_i}^{N,\mathcal{P}_\theta}$, $\hat{\bar{\mu}}_{t_i} := \hat{\bar{\mu}}_{t_i}^{N,\mathcal{P}_\theta}$, $\hat{\mu} := \hat{\mu}^N$.

In fact, we use the Minimal Wasserstein distance estimator, as the properties of this estimator have been outlined in (Bernton et al. (2017)). We apply it to our specific problem to control the structural loss. We first introduce the following assumptions:

**Assumption 2.2.** *The model is identifiable: there exists a unique $\mathcal{P}_\theta^* \in \Theta$ such that $\mathcal{W}_2(\bar{\mu}_T^{\mathcal{P}_\theta^*}, \nu) = \mathcal{W}_2(\mu^*, \nu)$ for every $\nu$ and*

$$\mathcal{P}_\theta^* = \arg\min_{\mathcal{P}_\theta \in \Theta} \mathcal{W}_2(\mu^*, \bar{\mu}_T^{\mathcal{P}_\theta}).$$

This assumption ensures the existence of a deterministic parameter in the SDE.

**Assumption 2.3.** *Data processes are sufficient: The data generation process error is satisfied $\mathcal{W}_2(\hat{\mu}^N, \mu^*) \to 0$, $\mathbb{P}$-almost surely, as $N \to \infty$.*

**Assumption 2.4.** *Continuity: The map $\mathcal{P}_\theta \mapsto \bar{\mu}_T^{\mathcal{P}_\theta}$ is continuous in the sense that $D(\mathcal{P}_\theta^N, \mathcal{P}_\theta) \to 0$ implies $\bar{\mu}_T^{\mathcal{P}_\theta^N} \overset{w}{\to} \bar{\mu}_T^{\mathcal{P}_\theta}$ as $N \to \infty$. $D^1$ is the metric of the parameter.*

**Assumption 2.5.** *Level boundedness: For some $\epsilon > 0$*
*The set $B(\epsilon) = \left\{ \mathcal{P}_\theta \in \Theta : \mathcal{W}_2(\mu^*, \bar{\mu}_T^{\mathcal{P}_\theta}) \leq \inf_{\theta \in \Theta} \mathcal{W}_2(\mu^*, \bar{\mu}_T^{\mathcal{P}_\theta}) + \epsilon \right\}$ is bounded.*

**Theorem 2.6.** *Consistency of the structural loss. Assuming that 2.2,2.3,2.4 and 2.5 hold, the loss function in (5) $\mathcal{W}_2(\hat{\bar{\mu}}_T^{N,\mathcal{P}_\theta}, \hat{\mu}^N) \leq \epsilon_l$ where $\epsilon_l \to 0$, $\mathbb{P}$-almost surely. Then there exists $aE \subset \Omega$ with $P(E) = 1$ such that for all $\omega \in E$ :*

$$\inf_{\mathcal{P}_\theta \in \Theta} \mathcal{W}_2(\hat{\mu}^N(\omega), \bar{\mu}_T^{\mathcal{P}_\theta}) \to \inf_{\mathcal{P}_\theta \in \Theta} \mathcal{W}_2(\mu^*, \bar{\mu}_T^{\mathcal{P}_\theta}), \tag{8}$$

*and there exists $n(\omega)$ such that,for all $N \geq n(\omega)$*

$$\mathcal{P}_\theta^N \to \mathcal{P}_\theta^* \text{ as } N \to \infty, \epsilon_l \to 0, \mathbb{P}\text{-almost surely.} \tag{9}$$

*Proof.* The proof is based on (Bernton et al. (2017)). However, the key difference is that we introduced a loss function control term, which enhances the result and more precise in appendix 8.3. $\quad\square$

After introducing the theorem above, we provide an error estimation for the diffusion bridge model.

**Theorem 2.7.** *Consistency of the diffusion bridge model: Assuming that 2.2,2.3,2.4 and 2.5 hold, the loss function in (5) satisfied $\mathcal{W}_2(\hat{\bar{\mu}}_T^{N,\mathcal{P}_\theta}, \hat{\mu}^N) \leq \epsilon_l$ where $\epsilon_l \to 0,$, $\mathbb{P}$-almost surely for $\mathcal{P}_\theta$.*

$$\mathcal{W}_2(\mu_T^{\mathcal{P}_\theta}, \mu^*) \to 0 \text{ as } N \to \infty, \epsilon_l \to 0, \mathbb{P}\text{-almost surely.}$$

---

[1] $D(P_\theta, P_\eta) = d_x(X_{0,\theta_3}, X_{0,\eta_3}) + d_b(b_{\theta_1}, b_{\eta_1}) + d_\sigma(\sigma_{\theta_2}, \sigma_{\eta_2})$, where the $d_x, d_b, d_\sigma$ correspond to distances in the appropriate space.

*Proof.* This result can be proved by combining Theorem 2.1 with Theorem 2.6. $\qquad\square$

**Know the target distribution:** This scenario has been extensively studied using MCMC algorithms and SDE-type samplers. Our method can still be applied to match a diffusion bridge, utilizing two primary matching methods. The first method involves specifying a density function, using existing MCMC algorithms to obtain $N$ discrete points at each position $X_t$, and then employing the Wasserstein distance loss as described above for matching.

**Note:** The design of the loss function is not unique. The diffusion bridge matching method presented here serves as a baseline algorithm that can be replaced by many other algorithms. We employ a Neural SDE bridge for the following reasons: (1) We aim to minimize the number of steps to reach the target distribution within the smallest possible time interval to reduce the amount of PINN training. (2) In cases with only partially observed samples, where density information is absent, the matching process is not unique and relies on the chosen model. (3) We simulate an equal number of Brownian motion paths and use a fully trainable initial value for drift and diffusion coefficients to ensure maximum flexibility. The Wasserstein distance guarantees the stability of training and overall match between the generated samples and the target value.

In practical scenarios, the maximum number of training points for the PINN is $MN$, where $M$ is the number of iterations of the SDE solver, satisfying $(M-1)h = T$. Given $\varepsilon$ and $M_0$, we aim to achieve $M \leq M_0$ by choosing appropriate $h$ and $T$ such that the following conditions hold simultaneously:

$$[\frac{T}{h}] \leq M_0 \quad \text{and} \quad Ch^{\frac{1}{2}} \exp\left(4L^2 T\right) \leq \varepsilon. \tag{10}$$

This is relatively easy to achieve because we have parameterized the initial values, allowing us to control them and, consequently, control $C$. This approach is distinctive and innovative compared to other diffusion bridge models. Additionally, this method can also match the Markov chain in cases where the density is known. The first one involves specifying a density function and then using existing MCMC algorithms to obtain $N$ discrete points at each position $X_t$. We then employ the same Wasserstein distance loss (6) as mentioned above for matching. The second method involves using transition density matching in (Dietrich et al. (2023)). Specifically, given a density function $f$, we can determine a transition density function $p(y|x, h)$ in MCMC algorithms. Then, by discretizing the SDE using the Euler-Maruyama method, we obtain the following transition density:

$$\hat{p}(y|x, h) = \mathcal{N}(y; x + b_{\theta_1}(x, t)h, h\sigma_{\theta_2}(x, t)\sigma_{\theta_2}^T(x, t)). \tag{11}$$

We can consider the following loss function:

$$\mathcal{P}_\theta{}^* = \arg\min_{\mathcal{P}_\theta \in \Theta} \iint [\hat{p}(y|x, h) - p(y|x, h)]^2 dy dx + [X_0 - X_{0,\theta_3}]^2. \tag{12}$$

Using an SDE-type sampler directly as the diffusion bridge is also feasible, eliminating the need for a matching process. A straightforward method for this purpose is the Langevin diffusion. Our experiments demonstrate the improved estimates provided by FKEE for the Langevin diffusion equation. The parameter pairs determining the diffusion bridge are $(X_0, b, \sigma, T)$. Many MCMC algorithms can be reduced to a diffusion bridge model, as shown in Table 1 in Appendix 8.1. Some of the more representative recent works include (Song & Ermon (2019)) for the case where $P$ is unknown, and (Vargas et al. (2023)) and (Grenioux et al. (2024)) for the case where $P$ is known.

### 2.3 FEYNMAN-KAC MODEL

This section presents our main contribution: a novel approach to expectation estimation. We aim to estimate $\mathbb{E}_{X \sim P}[f(X)]$ by decoding $P$. All relevant information about $P$ is encapsulated in $(X_0, b, \sigma, T)$. The decoding loss measures the accuracy of our estimate, while the decoding speed impacts the algorithm's efficiency. Our key innovation is the direct utilization of information within $(X_0, b, \sigma, T)$, as it contains all the necessary information about $P$. This is the core of our algorithm. The decoding process can be viewed as an approximation of the Feynman-Kac operator, formally obtained by solving the Feynman-Kac equation.

The Feynman-Kac operator (Del Moral & Del Moral (2004)) is crucial in translating between deterministic PDEs and stochastic processes through the Feynman-Kac formula (Feynman-Kac equation). The Feynman-Kac equation (Pham (2014)) is a powerful method for solving PDEs by linking them to stochastic processes. The basic idea is to represent the solution of a PDE as the expectation of a function of a stochastic process and use Monte Carlo methods to approximate this expectation.

In our approach, we can reverse the process to obtain new methods for deriving MCMC results. Specifically, we can use the solution of a PDE to accurately express the corresponding MCMC results. We consider the following simplified version of the Feynman-Kac formula, which is commonly encountered.

**Theorem 2.8.** *Feynman-Kac formula: Assuming that SDE (1) has strong solutions and $f$ is a function in $C^2$, we have the following Feynman-Kac equation, which has unique solutions on the interval $[0, T]$.*

$$\frac{\partial u(x,t)}{\partial t} + \sum_{i=1}^{d} \frac{\partial u(x,t)}{\partial x_i} b_i(x,t) + \frac{1}{2} \sum_{i=1}^{d} \sum_{j=1}^{d} \frac{\partial^2 u(x,t)}{\partial x_i \partial x_j} (\sigma(x,t)\sigma(x,t)^T)_{i,j} = 0,$$

*with the boundary conditions*

$$u(x,T) = f(x).$$

*The solution to the Feynman-Kac equation at the initial time is $u(x_0,0) = \mathbb{E}[f(X_T)|X_0 = x_0]$.*

*Proof.* The proof of the theorem is a classical result. For more details, please refer to Särkkä & Solin (2019). □

**Fast calculation method.** Calculating this equation involves computing the Hessian matrix of a function and some partial derivatives, which can be obtained using any library with automatic differentiation, such as `Pytorch`. If we consider only the diagonal diffusion coefficients $\sigma$, the algorithm can be accelerated. For instance, in Langevin diffusion where $\sigma = I_{d \times d}$, we need to calculate the second-order partial derivatives of the main diagonal.

For the Neural SDE, to reduce computation, we can consider a diagonal diffusion matrix function $\sigma : \mathbb{R}^d \times [0, T] \to \Lambda(\mathbb{R}^d)$, where $\Lambda(\mathbb{R}^d)$ is the set of real-valued diagonal matrices. We only calculate the second-order derivatives of the diagonal elements to avoid the entire diffusion matrix function. To achieve this, we design the following loss functions:

$$\mathcal{L}_1 = \iint_{\mathcal{D} \times [0,T]} \left[ \frac{\partial u_\theta(x,t)}{\partial t} + \sum_{i=1}^{d} \frac{\partial u_\theta(x,t)}{\partial x_i} b_i(x_t,t) + \frac{1}{2} \sum_{i=1}^{d} \frac{\partial^2 u_\theta(x,t)}{\partial x_i^2} \text{diag}(\sigma^2(x,t))_i \right]^2 dx \, dt \tag{13}$$

and

$$\mathcal{L}_2 = \int_{\mathcal{D}} [u_\theta(x,T) - f(x)]^2 \, dx. \tag{14}$$

Finally, we obtain the solution by optimizing these two loss functions.

$$u^*(x,t) = \underset{u_\theta(x,t)}{\arg\min} \left[ \lambda_1 \mathcal{L}_1 + \lambda_2 \mathcal{L}_2 \right],$$

where $\lambda_1$ and $\lambda_2$ are the weights of the two loss functions. $u_\theta(x,t)$ is the neural network with a *tanh* activation function. Ultimately, we can obtain the expectation $u^*(x_0,0) = \mathbb{E}[f(X_T)|X_0 = x_0]$. The term $\mathcal{L}_d = \lambda_1 \mathcal{L}_1 + \lambda_2 \mathcal{L}_2$ represents the empirical decoding loss, which incurs statistical and optimization errors compared to the true decoding loss. Error analysis for this equation can be found in many works related to PINN, for example, in (De Ryck & Mishra (2022)). Algorithm 2 presents a simplified version, while the detailed implementation of the algorithms can be found in Appendix 10. In the approximation, we sample the PDE domain. In Figure 1, we simulate the SDE trajectory and compute the PINN loss at these positions, which differs from directly using the SDE endpoint, as our expectation comes from initial moment of solution.

**Viewing MCMC expectation estimators from a decoding perspective** The decoding loss of traditional MCMC expectation estimators might be suboptimal because these estimators often do not fully utilize the information about $(X_0, b, \sigma, T)$. Typically, these estimators rely on simulating a

---

**Algorithm 2** Feynman-Kac model (FCM)

---

**Input**: Points of observation: $X_t$, Drift coefficient: $b(t, X_t)$, Diffusion coefficient: $\sigma(t, X_t)$. Neural network: $u_\theta(x, t)$. The function $f$. Required error threshold $\varepsilon$.
**Output**: $\mathbb{E}(f(X_T)|X_0 = x_{t_0}) = u_\theta(x_{t_0}, t_0)$

1: Calculate PDE loss $\mathcal{L}_1$ in (13).
2: Calculate boundary loss $\mathcal{L}_2$ in (14).
3: Update parameters $\theta$.
4: **if** $(\lambda_1 \mathcal{L}_1 + \lambda_2 \mathcal{L}_2) < \varepsilon$ **then**
5:     End of training
6: **end if**

---

subset of samples for averaging, which can introduce local bias and fail to provide a comprehensive estimate of the entire distribution. The method of control functions (Oates et al. (2014); South et al. (2020)) attempts to mitigate this by reusing information from $P$, but it is not universally applicable to any $P$ and $f$. Additionally, these methods require stronger assumptions to guarantee accurate estimates, influenced by LLN and the ETMC, which can further reduce the efficiency of MCMC algorithms. Therefore, traditional MCMC expectation estimators can be seen as incomplete decoding.

**Discussion of the choice of the Feynman-Kac model** Our approach fundamentally changes how expectations are calculated by utilizing the full distributional information in $P$ to approximate $(X_0^*, b^*, \sigma^*)$. However, when approximating $(X_0^*, b^*, \sigma^*)$ in many diffusion bridge models, it is often necessary to simulate part of the Brownian motion trajectory to estimate the loss function. This results in some positions $(x, t)$ corresponding to $(b, \sigma)$ being accurate, while others depend on the network's generalization ability. Consequently, the appearance of $x$ in our position $(x, t)$ occurs randomly, necessitating a meshless PDE solver.

For certain $(b, \sigma)$ with exact analytical forms and diffusion bridges that exhibit better generalization, a non-meshless PDE solver may suffice. The second critical issue is the change in how expectations are computed, introducing the dimension $d$ with respect to the MCMC expectation estimator. To overcome the curse of dimensionality, we need a PDE solver capable of handling this problem. For low-dimensional, non-meshless scenarios, finite element methods (Milstein et al. (2004)) are viable. However, in more general cases, we require meshless PDE solvers that can address the curse of dimensionality. We have chosen a classical PDE solver called PINN, but other PDE solvers meeting these conditions are also feasible.

### 2.4 Feynman-Kac Operator Expectation Estimator

FKEE consists of two parts: the Diffusion Bridge Model and the Feynman-Kac Model. The Diffusion Bridge Model provides the coefficients and initial values of the SDE for the Feynman-Kac Model. For a target distribution, we first use the Diffusion Bridge Model to obtain the corresponding coefficients and save them. The Feynman-Kac Model then uses these coefficients to directly approximate $\mathbb{E}_{X \sim P}[f(X)]$. Since the Feynman-Kac Model is trained using PINN, we can leverage GPU arithmetic acceleration or parallelism to efficiently handle high-dimensional distributions and obtain the corresponding results.

## 3 Discussion

**Discussions on $P$:** In conventional MCMC algorithms like Langevin diffusion, extensive analysis is conducted on the properties of potential energy functions, particularly the requirements for Lipschitz continuous gradients and strong convexity, as detailed in (Cheng & Bartlett (2018); Cheng et al. (2018)). However, our approach diverges by not depending on these specific properties of energy functions for convergence and speed. Instead, we require the corresponding SDE to have strong solutions and the Feynman-Kac equation to be well defined. To better understand the applicability of our method, consider the Itô-type SDE (1), which corresponds to the Fokker–Planck–Kolmogorov

(FPK) equation (Risken & Risken (1996); Frank (2005)):

$$\frac{\partial p(x,t)}{\partial t} = -\sum_i \frac{\partial}{\partial x_i}[b_i(x,t)p(x,t)] + \frac{1}{2}\sum_{i,j}\frac{\partial^2}{\partial x_i \partial x_j}\left\{\left[\sigma(x,t)\sigma^\top(x,t)\right]_{ij}p(x,t)\right\},$$

where $p(x,t)$ is the probability density function of $X_t$. For the stationary distribution, we set $\frac{\partial p(x,t)}{\partial t} = 0$. There can be multiple pairs $(b,\sigma)$ that satisfy this stationary FPK equation, with Langevin diffusion being a special case where $\sigma = I_d$. Our method can handle various other cases as well, such as those discussed in (Li (2023)). Additionally, our method applies to more general pairs $(b,\sigma)$ that satisfy the FPK equation, even under finite time and non-stationary conditions.

**Discussion on $f$:** In classical MCMC expectation estimators, the following computation forms are used:

$$\mathbb{E}\left[f(X)\right] = \frac{1}{N}\sum_{i=1}^{N}f(X_T^i), \tag{15}$$

where $X_T^i$ is the value at moment $T$ of the $i$th Markov chain, with different Markov chains being independent. This represents the classical Monte Carlo integral calculation, where error is based on the LLN. Another estimator is applicable only when $P$ is the stationary distribution:

$$\mathbb{E}\left[f(X)\right] = \frac{1}{N-M}\sum_{t=M}^{N}f(X_t). \tag{16}$$

Here, averaging is done over the time span of a Markov chain, with $M$ denoting the number of samples discarded during the burn-in period, characterized by correlated samples. Error in this case is influenced by the ETMC, making optimal $M$ selection challenging for complex problems. Incorporating relevant samples can reduce the impact on MCMC expectation estimator efficiency. In difficult scenarios, properties of $f$ can often lead to larger biases. Our approach enhances MCMC efficiency by utilizing points within the burn-in period for PDE loss computation and refining assumptions on $f$, offering a novel bias reduction method. In summary, one method relies on the LLN, often requiring Lipschitz continuity of $f$, with the estimator's variance related to the Lipschitz coefficient of $f$. The other is based on the ETMC, also imposing requirements on the Lipschitz coefficient and the density function of $P$. Our method, however, only requires that the boundary conditions of the PDE satisfy a specific smoothness, namely $f \in C^2$. This significantly broadens the scope of this approach.

## 4 EXPERIMENTS

**Partition Function Computation for Random Graph Models.** In our first example, we focus on computing the partition function for random graph models, simplifying the setup from (Haddadan et al. (2021)) to estimate the mathematical expectation and the corresponding partition function. Background and details can be found in Appendix 9.1. Specifically, we consider the estimation of the matching function of the Ising model for the high temperature case, i.e., the case corresponding to a smaller $\beta$. we aim to estimate the following two expectation:

$$\mathbb{E}F = \mathbb{E}\exp(-\frac{\beta_2 - \beta_1}{2}H(X_{\beta_1})), \mathbb{E}G = \mathbb{E}\exp(\frac{\beta_2 - \beta_1}{2}H(X_{\beta_2})),$$

where $X_{\beta_i}$ is a Gibbs distribution of the 2D Ising model (The dimension is $n \times n$) with parameter $\beta_i$, which is given in advance. The MCMC expectation estimators may exhibit considerable bias due to the involvement of the exponential function, the discrete nature of the target distribution. In addition, Gibbs samplers require a large mixing time to reach the stationary distribution in low-temperature regimes. We compare three different methods: the first is the SOTA MCMC expectation estimator, the second uses the diffusion bridge model to resample data for averaging, and the third combines the diffusion bridge model with the Feynman-Kac model, also known as the FKEE. The estimators are denoted as MCMC-C, MCMC-R, and MCMC-T. $wi, vi, q$ represent the estimated values of the corresponding $\mathbb{E}F, \mathbb{E}G, Q = \frac{\mathbb{E}F}{\mathbb{E}G}$. We record the number of points sampled from the Gibbs chain and the algorithm's total runtime, excluding the mixing time. The dataset is obtained from a Gibbs chain that has already reached its stationary distribution. We consider two computational methods

based on the boundary conditions. The first treats $H(X_\beta)$ as $Y_T$, resulting in a one-dimensional approximate distribution with boundary condition $\exp\{-\beta Y\}$. The second uses $X_\beta$ as $Y_T$, leading to a high-dimensional case with boundary condition $\exp\{-\beta H(Y_T)\}$. Figure 2 shows the logarithmic squared errors of the two estimators, $wi$ and $vi$, for various methods. Note that for $n > 5$, MCMC-C fails to provide stable estimates due to excessive computational costs (Haddadan et al. (2021)). The training samples we used are the same as those used in MCMC-C (only differing in numbers). MCMC-R can be regarded as the result of using the diffusion bridge model to generate the same number of samples and then averaging the results. It can be observed from the figure that FKEE (MCMC-T) leverages the diffusion bridge model more effectively, providing better results, even in high-dimensional scenarios. Figure 3 evaluates the efficiency. Since the same sample size is used for computing $wi$ and $vi$, this part is omitted. It can be seen that the sample size used by MCMC-C is significantly larger than that of FKEE. Regarding time costs, MCMC-R records the training time of the diffusion bridge, while FKEE records the total training time of the entire algorithm (including both the diffusion bridge and the FK model). It can be observed that even in high-dimensional scenarios, the computations can still be completed within an acceptable time. This experiment highlights the effect of f on the target distribution's expectation and the algorithm's efficiency, defined here as **using fewer points on the Markov chain to achieve higher accuracy in approximating expectations.**

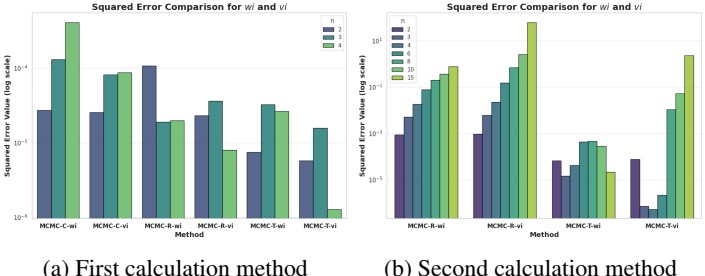

(a) First calculation method        (b) Second calculation method

Figure 2: Comparison of squared errors

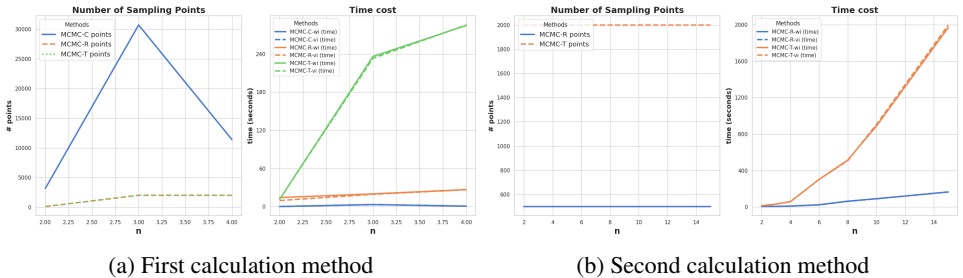

(a) First calculation method        (b) Second calculation method

Figure 3: Comparison of time and number of points sampled

**Baseline Experiments on Properties of $P$ and Low Variance.** Additional baseline experiments on the properties of $P$ and low variance are presented in Appendix 9.3, where we simulate the same trajectory using the Langevin diffusion model with various expectation computation methods, resulting in different estimations. In Appendix 9.2, we evaluate the proposed diffusion bridge model by generating samples and compare the distributions of the initial and subsequent sample sets, showcasing the effectiveness of the diffusion bridge model.

## 5 CONCLUSION

We introduce a heuristic method for estimating mathematical expectations by bridging the gap between deep learning PDE solvers and sampling methods. This approach reduces reliance on traditional assumptions (LLN and ETMC) and expands the applicability to a broader range of $P$ and $f$. We presented a versatile diffusion bridge model to extend the range of $P$ and utilized PDE methods to broaden the scope of $f$. Our method demonstrates potential significance across multiple domains.

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

# 6 APPENDIX

# 7 RELATED WORK

The diffusion model belongs to a class of stochastic differential equations, which are used to approximate the target distribution. It has been widely used for generative models (Dhariwal & Nichol (2021)), variational inference (Geffner & Domke (2021); Kingma et al. (2021)), etc. The diffusion bridge model is a variant of the diffusion model. Early development of diffusion bridge models involved simulating processes originating from two endpoints (Beskos et al. (2008)). Alternative approaches for constructing diffusion bridge models are outlined by (Liu et al. (2022); Bladt & Sørensen (2010)). Since the diffusion bridge model essentially functions as a sampling algorithm, it plays a pivotal role in addressing the crucial task of high-dimensional distribution sampling. Sampling high-dimensional distributions is a fundamental task with applications across various fields. Common methods include MCMC, random flow, and generative models. Recent work includes stream-based methods (Müller et al. (2018); Yang et al. (2017); Matsubara et al. (2020); Strathmann et al. (2015); Tran et al. (2019)), MCMC-based methods (Deng et al. (2020); Chen et al. (2014); Jacob et al. (2017)) and generative models (Nichol & Dhariwal (2021)), score-based models (Song et al. (2020); Song & Ermon (2019)). Normalizing Flows (Albergo & Vanden-Eijnden (2022)). These models can be broadly categorized into two groups: those based on given discrete points and those relying on a given density function. The former primarily serves for learning and generating real world data such as text and images, while the latter is used for sampling, statistical estimation, and similar purposes. Notably, Langevin diffusion (Cheng et al. (2018); Xifara et al. (2013); García-Portugués et al. (2017)) is a classical model within the latter category. (Zhang (2024)) presents an explicit construction of the drift coefficient for two scenarios: when $P$ has a closed-form expression and when it does not.

The Feynman-Kac model is a technique employed to solve partial differential equations (PDEs) by using deep learning. Deep learning has found application in solving PDEs of the Feynman-Kac equation type, as demonstrated by (Berner et al. (2020); Blechschmidt & Ernst (2021). Liang & Borovkov (2023)) highlights the approximation of Feynman-Kac type expectations through the approximation of discrete Markov chains, thereby enhancing the order of convergence. When employing PINN to solve Feynman-Kac type PDEs, the sampling algorithm can be linked to the path of the SDE. This approach enables the acquisition of adaptive sampling points from the paths of SDE, which proves more efficient than uniform point selection (Chen et al. (2023)). Further analysis of the approximation error for this class of equations is presented in (De Ryck & Mishra (2022)).

# 8 DEFINITIONS AND RELATED THEORY

## 8.1 COMPARISON OF DIFFUSION BRIDGE MODEL.

## 8.2 DEFINITIONS

**Wasserstein distance:** The most commonly used measure of distance between probability distributions is the Wasserstein distance. It calculates the minimum cost of transporting mass from one

Table 1: Comparison of Diffusion bridge model

| Method | $X_0$ | $b$ | $\sigma$ | $T$ | Descriptions |
|---|---|---|---|---|---|
| Classical MCMC | $\forall \mathbf{x}_0 \in \mathbb{R}^d$ | $p(y\|x)$ | $p(y\|x)$ | $\infty$ | $p(y\|x)$ is the transfer probability density function in the MCMC algorithm. The meaning of $p(y\|x)$ is that the corresponding coefficients can be obtained by a SDE. |
| Langevin MCMC | $\forall \mathbf{x}_0 \in \mathbb{R}^d$ or $\forall X_0 \sim P_0$ | $\frac{1}{2}\nabla_x \log p(x)$ | $I_{d \times d}$ | $\infty$ | $p(x)$ is target density function and density function of a stationary distribution. $P_0$ is the initial distribution. |
| Score-based SDE and diffusion models (DDPM) | $\forall X_0 \sim P_0$ | $f(\mathbf{x},t) - g^2(t) \nabla_{\mathbf{x}} \log p_t(\mathbf{x})$ | $g(t)$ | $\infty$ | $\nabla_{\mathbf{x}} \log p_t(\mathbf{x})$ is obtained from the data and $f(x,t)$ and $g(t)$ are known. $P_0$ is the prior distribution. |
| Flow match ODE | $\forall X_0 \sim P_0$ | $v(x,t)$ | $0$ | $1$ | $v(x,t)$ is obtained by matching the data. $P_0$ is the initial distribution. |
| Neural SDE bridge **(taken in this paper)** | $x_0 = x_{0,\theta_3}$ | $b_{\theta_1}(x,t)$ | $\sigma_{\theta_2}(x,t)$ | $< \infty$ | $b_{\theta_1}(x,t)$ and $\sigma_{\theta_2}(x,t)$ is obtained from the data or match method. |

distribution to another, based on the distance between the points being transported and the amount of mass being moved. The Wasserstein distance is especially beneficial for comparing distributions with different shapes since it considers the structure of distributions instead of only their statistical moments. This distance metric is widely applied in fields like image processing, computer vision, and machine learning. The definition is

$$\mathcal{W}_p(\mu,\nu) := \left( \inf_{\pi \in \Pi(\mu,\nu)} \int_{\mathbb{R}^d \times \mathbb{R}^d} |x-y|^p \pi(dx,dy) \right)^{\frac{1}{p}} = \inf \left\{ \left[ \mathbb{E}|X-Y|^p \right]^{\frac{1}{p}}, \mathbb{P}_x = \mu, \mathbb{P}_Y = \nu \right\}.$$

$\Pi(\mu,\nu)$ denotes the class of measures on $\mathbb{R}^d \times \mathbb{R}^d$ with marginal distributions $\mu$ and $\nu$.

**Euler-Maruyama method:** (Platen (1999)) is a frequently used approach for solving SDE through an iterative format. This method has been shown to converge to a strong order of $\mathcal{O}(h^{\frac{1}{2}})$, where the error is dependent on the Lipschitz coefficients of the drift and diffusion coefficients. When generating paths using this method, it is recommended to use smaller step sizes, to minimize the errors associated with the method.

$$X_{t+h} = X_t + b(X_t,t)h + \sigma(X_t,t)(W_{t+h} - W_t), X_0 = x_0.$$

Numerical solvers for stochastic differential equations of any accuracy are allowed when constructing sample paths for diffusion.

**Physics-informed neural networks:** PINN (Raissi et al. (2019)) is a deep learning method for solving partial differential equations. The main idea is to use neural networks for fitting solutions to PDE problems, PINN incorporates the residuals of the PDE (the difference between the left-hand side and the right-hand side of the PDE equation) into the loss function, and then updates the weights and parameters of the neural network through a backpropagation algorithm. Specifically, we consider follow PDE:

$$F(u_t, u_x, u_{xx}) = g(u,x,t)$$

and the boundary condition is

$$G(u_t, u_x, u_{xx}) = 0$$

We choose a neural network $u^\theta(x,t)$ to approximate the solution $u(x,t)$. By automatic differentiation, we can easily obtain the term $u_t^\theta, u_x^\theta$ and $u_{xx}^\theta$. We then need to sample the region of the target and calculate the value of the empirical loss function for these points. Finally the solution $u_t^\theta$ is obtained by optimising the combination of the two loss functions.

$$\textit{Loss PDE} = F(u_t^\theta, u_x^\theta, u_{xx}^\theta) - g(u^\theta,x,t) \quad \textit{Loss boundary} = G(u_t^\theta, u_x^\theta, u_{xx}^\theta)$$

$$\textit{Loss} = \lambda_1 \textit{Loss boundary} + \lambda_2 \textit{Loss PDE}$$

$\lambda_1$ and $\lambda_2$ are the weights of the two loss functions.

## 8.3 PROOF OF THEORETICAL RESULTS

**Theorem 8.1.** *If the drift and diffusion coefficients satisfy the conditions (Platen (1999)) in SDE (1):*

- *Lipschitz condition*

$$|b(x,t) - b(y,t)| \leq K|x - y| \ and \ ||\sigma(x,t) - \sigma(y,t)||_F \leq K|x - y|$$

  *for all $t \in [0, T]$ and $x, y \in \mathbb{R}^d$*

- *Linear growth bound*

  *There exists a constant $C$ such that*

$$|b(x,t)|^2 \leq C^2(1 + |x|^2) \quad and \quad ||\sigma(x,t)||_F^2 \leq C^2(1 + |x|^2)$$

  *for all $t \in [0, T]$ and $x, y \in \mathbb{R}^d$.*

- *Measurability*
$$b(x,t) \ and \ \sigma(x,t) \ is \ jointly \ measurable.$$

- *Initial value*
$$X_0 \ is \ \mathcal{F}_0\text{-}measurable \ with \ \mathbb{E}(|X_0|^2) < \infty.$$

*where $|| \cdot ||_F$ denotes the Frobenius norm of a matrix. Then the SDE has a unique strong solution. The solution can be controlled by the initial value, i.e., $\mathbb{E}X_T^2 \leq C\mathbb{E}X_0^2$.*

**Proof for Theorem 2.1.**

*Proof.* Base on the Lipschitz condition, Linear growth bound condition in Theorem 8.1, and the Itô isometry, we can derive the following:

$$\mathcal{W}_2^2(\mu_T, \bar{\mu}_T) \leq \mathbb{E}|X_T - \bar{X}_T|^2$$

$$\leq 4\mathbb{E}\left|\int_0^T b(X_t, t) - b(\bar{X}_t, t)dt\right|^2 + 4\mathbb{E}\left|\int_0^T \sigma(X_t, t) - \sigma(\bar{X}_t, t)dW_t\right|^2.$$

$$+ 4\mathbb{E}\left|\int_{[T]}^T \sigma(\bar{X}_t, t)dW_t\right|^2 + 4\mathbb{E}\left|\int_{[T]}^T b(\bar{X}_t, t)dt\right|^2.$$

$$\leq 4L^2\mathbb{E}\int_0^T |X_t - \bar{X}_t|^2dt + 4L^2\mathbb{E}\int_0^T |X_t - \bar{X}_t|^2dt + 4\mathbb{E}\left|\int_{[T]}^T \sigma(\bar{X}_t, t)dW_t\right|^2 + 4\mathbb{E}\left|\int_{[T]}^T b(\bar{X}_t, t)dt\right|^2.$$

$$\leq 8L^2\mathbb{E}\int_0^T |X_t - \bar{X}_t|^2dt + 4C^2\mathbb{E}\left|\int_{[T]}^T \bar{X}_t^2dt\right| + 4C^2\mathbb{E}\left|\int_{[T]}^T \bar{X}_tdt\right|^2 + 4C^2h^2.$$

$$\leq 8L^2\mathbb{E}\int_0^T |X_t - \bar{X}_t|^2dt + 4C^2M_0h + 4C^2M_1h + 4C^2h.$$

where $M_0$ and $M_1$ are upper bounds that relevant to $\mathbb{E}(|\bar{X}_T|^2)$ because of $\mathbb{E}(|X_0|^2) < \infty$. $[T] := \max\{Mh \leq T\}$. Finally, based on the Gronwall's inequality.

$$\mathcal{W}_2^2(\mu_T, \bar{\mu}_T) \leq \mathbb{E}|X_T - \bar{X}_T|^2 \leq Ch\exp(8L^2T).$$

$\square$

**Proof for Theorem 2.6.**

Before proving this theorem we introduce the following definitions and lemmas. Some of these definitions and lemmas are taken from (Rockafellar & Wets (2009), Fournier & Guillin (2015))

**Definition 8.2.** *The function $f : \Theta \to \mathbb{R} :$ is lower semicontinuous at $x_0$ if*

$$\liminf_{x \to x_0} f(x) \geq f(x_0). \tag{17}$$

**Definition 8.3.** *A sequence of functions $f_n : \Theta \to \mathbb{R}$ is said to epi-converge to $f : \Theta \to \mathbb{R}$ if for all $\theta \in \Theta$*

$$\begin{cases} \liminf_{n \to \infty} f_n(\theta_n) \geq f(\theta) & \text{for every sequence } \theta_n \to \theta, \\ \limsup_{n \to \infty} f_n(\theta_n) \leq f(\theta) & \text{for some sequence } \theta_n \to \theta. \end{cases} \tag{18}$$

**Lemma 8.4.** *The sequence $f_n : \Theta \to \mathbb{R}$ epi-converges to $f : \Theta \to \mathbb{R}$ if and only if*

$$\begin{cases} \liminf_{n \to \infty} \inf_{\theta \in \mathcal{K}} f_n(\theta) \geq \inf_{\theta \in \mathcal{K}} f(\theta) & \text{for every compact set } \mathcal{K} \subset \Theta, \\ \limsup_{n \to \infty} \inf_{\theta \in \mathcal{O}} f_n(\theta) \leq \inf_{\theta \in \mathcal{O}} f(\theta) & \text{for every open set } \mathcal{O} \subset \Theta. \end{cases} \tag{19}$$

**Lemma 8.5.** *Varadarajan's theorem: If $X_1, \ldots, X_n$ are i.i.d. $X \sim P$ on a separable metric space, then $\hat{\mu}^n \overset{w}{\to} \mu^*$ $\mathbb{P}$-almost surely where $\hat{\mu}^n$ is empirical measure.*

**Lemma 8.6.** *Attainment of a minimum: Suppose $f : \Theta \to \mathbb{R}$ is lower semicontinues, level-bounded and proper. Then the value $\inf f$ is finite and the set $\arg\min f$ is nonempty and compact.*

**Lemma 8.7.** *The properties of epi-convergence: If $f_1^n \leq f^n \leq f_2^n$ with $f_1^n \overset{\text{epi}}{\to} f$ and $f_2^n \overset{\text{epi}}{\to} f$, then $f^n \overset{\text{epi}}{\to} f$.*

**Lemma 8.8.** *The limits of inf: Suppose $f_n \overset{\text{epi}}{\to} f$ with $-\infty < \inf f < \infty$. Then $\inf f^n \to \inf f$ if and only if there exists for every $\varepsilon > 0$ a compact set $B \subset R^n$ along with an index set $\mathcal{N}$ such that*

$$\inf_B f^n \leq \inf f^n + \varepsilon \quad \text{for all } n \in \mathcal{N}.$$

*Proof.* Based on Assumptions 2.4 and Villani (2008). we can conclude the map is lower semicontinuous. i,e, $\mathcal{W}_2(\bar{\mu}_T^{\mathcal{P}_\theta}, \nu) \leq \liminf_{N \to \infty} \mathcal{W}_2(\bar{\mu}_T^{\mathcal{P}_\theta^N}, \nu)$.

Then, we aim to prove $\mathcal{P}_\theta \mapsto \mathcal{W}_2(\hat{\mu}^N, \bar{\mu}_T^{\mathcal{P}_\theta})$ epi converges to $\mathcal{P}_\theta \mapsto \mathcal{W}_2(\mu^*, \bar{\mu}_T^{\mathcal{P}_\theta})$ $\mathbb{P}$-almost surely.

Firstly, we can observe the following inequality:

$$\mathcal{W}_2(\mu^*, \bar{\mu}_T^{\mathcal{P}_\theta}) - \mathcal{W}_2(\mu^*, \hat{\mu}_T^N) \leq \mathcal{W}_2(\hat{\mu}_T^N, \bar{\mu}_T^{\mathcal{P}_\theta}), \tag{20}$$

and

$$\mathcal{W}_2(\hat{\mu}_T^N, \bar{\mu}_T^{\mathcal{P}_\theta}) \leq \mathcal{W}_2(\hat{\mu}_T^N, \hat{\bar{\mu}}_T^{N,\mathcal{P}_\theta}) + \mathcal{W}_2(\hat{\bar{\mu}}_T^{N,\mathcal{P}_\theta}, \bar{\mu}_T^{\mathcal{P}_\theta}). \tag{21}$$

In inequality (20), the function $\mathcal{P}_\theta \mapsto \mathcal{W}_2(\mu^*, \bar{\mu}_T^{\mathcal{P}_\theta})$ epi-converges to $\mathcal{P}_\theta \mapsto \mathcal{W}_2(\mu^*, \bar{\mu}_T^{\mathcal{P}_\theta})$ because it is independent of $N$, and the function $\mathcal{P}_\theta \mapsto \mathcal{W}_2(\mu^*, \hat{\mu}_T^N)$ epi-converges to $0$ as $N \to \infty$, due to Assumption 2.3.

In inequality (21), the function $\mathcal{P}_\theta \mapsto \mathcal{W}_2(\hat{\mu}_T^N, \hat{\bar{\mu}}_T^{N,\mathcal{P}_\theta})$ epi converge to $0$, because of assumptions about the loss function. Finally we aim to proof that the function $\mathcal{P}_\theta \mapsto \mathcal{W}_2(\hat{\bar{\mu}}_T^{N,\mathcal{P}_\theta}, \bar{\mu}_T^{\mathcal{P}_\theta})$ epi converge to $\mathcal{P}_\theta \mapsto \mathcal{W}_2(\mu^{\mathcal{P}_\theta^*}, \bar{\mu}_T^{\mathcal{P}_\theta})$. Combining Assumption 2.2 $\mathcal{W}_2(\mu^{\mathcal{P}_\theta^*}, \nu) = \mathcal{W}_2(\mu^*, \nu)$ and Lemma 8.7 we can conclude that $\mathcal{P}_\theta \mapsto \mathcal{W}_2(\hat{\mu}^N, \bar{\mu}_T^{\mathcal{P}_\theta})$ epi converges to $\mathcal{P}_\theta \mapsto \mathcal{W}_2(\mu^*, \bar{\mu}_T^{\mathcal{P}_\theta})$ $\mathbb{P}$-almost surely.

In the following we demonstrate that the function $\mathcal{P}_\theta \mapsto \mathcal{W}_2(\hat{\bar{\mu}}_T^{N,\mathcal{P}_\theta}, \bar{\mu}_T^{\mathcal{P}_\theta})$ epi converge to $\mathcal{P}_\theta \mapsto \mathcal{W}_2(\mu^{\mathcal{P}_\theta^*}, \bar{\mu}_T^{\mathcal{P}_\theta})$.

According to Lemma 8.4, we go on to verify two inequalities. For a compact set $\mathcal{K}$, by the lower semicontinuous of the map $\mathcal{P}_\theta \mapsto \mathcal{W}_2(\nu, \bar{\mu}_T^{\mathcal{P}_\theta})$, we have

$$\inf_{\mathcal{P}_\theta \in \mathcal{K}} \mathcal{W}_2(\hat{\bar{\mu}}_T^{N,\mathcal{P}_\theta^N}, \bar{\mu}_T^{\mathcal{P}_\theta}) = \mathcal{W}_2(\hat{\bar{\mu}}_T^{N,\mathcal{P}_\theta^N}, \bar{\mu}_T^{\mathcal{P}_\theta^N}), \text{ for some } \mathcal{P}_\theta^N \in \mathcal{K}. \tag{22}$$

$$\liminf_{N \to \infty} \inf_{\mathcal{P}_\theta \in \mathcal{K}} \mathcal{W}_2(\hat{\bar{\mu}}_T^{N,\mathcal{P}_\theta}, \bar{\mu}_T^{\mathcal{P}_\theta^N}) \overset{1}{=} \liminf_{N \to \infty} \mathcal{W}_2(\hat{\bar{\mu}}_T^{N,\mathcal{P}_\theta}, \bar{\mu}_T^{\mathcal{P}_\theta^N})$$

$$\overset{2}{=} \lim_{k \to \infty} \mathcal{W}_2(\hat{\bar{\mu}}_T^{N_k,\mathcal{P}_\theta}, \bar{\mu}_T^{\mathcal{P}_\theta^{N_k}}) \tag{23}$$

$$\overset{3}{=} \lim_{m \to \infty} \mathcal{W}_2(\hat{\bar{\mu}}_T^{N_{k_m},\mathcal{P}_\theta}, \bar{\mu}_T^{\mathcal{P}_\theta^{N_{k_m}}}),$$

$$\lim_{m\to\infty} \mathcal{W}_2(\hat{\bar{\mu}}_T^{N_{k_m},\mathcal{P}_\theta}, \bar{\mu}_\theta^{\mathcal{P}^{N_{k_m}}_\theta}) = \liminf_{m\to\infty} \mathcal{W}_2(\hat{\bar{\mu}}_T^{N_{k_m},\mathcal{P}_\theta}, \bar{\mu}_T^{\mathcal{P}^{N_{k_m}}_\theta})$$

$$\overset{4}{\geq} \mathcal{W}_2(\mu^{\mathcal{P}^*_\theta}, \bar{\mu}_T^{\mathcal{P}_{\theta_0}}) \tag{24}$$

$$\geq \inf_{\mathcal{P}_\theta \in \mathcal{K}} \mathcal{W}_2(\mu^{\mathcal{P}^*_\theta}, \bar{\mu}_T^{\mathcal{P}_\theta}),$$

where the "1" holds due to the substitution for the equation 22, "2" holds due to the definition of the infimum, "3" holds due to the compactness of the $\mathcal{K}$, "4" holds due to the Lemma 8.5 and lower semicontinuous.

For an open set $\mathcal{O}$,

$$\inf_{\mathcal{P}_\theta \in \mathcal{K}} \mathcal{W}_2(\hat{\bar{\mu}}_T^{N,\mathcal{P}_\theta}, \bar{\mu}_T^{\mathcal{P}_\theta}) \leq \mathcal{W}_2(\hat{\bar{\mu}}_T^{N,\mathcal{P}_\theta}, \bar{\mu}_\theta^{\mathcal{P}^N_\theta}), \ exist \ \mathcal{P}_\theta^N \in \mathcal{O}. \tag{25}$$

$$\limsup_{N\to\infty} \inf_{\mathcal{P}_\theta \in \mathcal{O}} \mathcal{W}_2(\hat{\bar{\mu}}_T^{N,\mathcal{P}_\theta}, \bar{\mu}_T^{\mathcal{P}_\theta}) \overset{1}{\leq} \limsup_{N\to\infty} \mathcal{W}_2(\hat{\bar{\mu}}_T^{N,\mathcal{P}_\theta}, \bar{\mu}_\theta^{\mathcal{P}^N_\theta})$$

$$\tag{26}$$

$$\overset{2}{\leq} \limsup_{N\to\infty} \mathcal{W}_2(\hat{\bar{\mu}}_T^{N,\mathcal{P}_\theta}, \mu^{\mathcal{P}^*_\theta}) + \limsup_{N\to\infty} \mathcal{W}_2(\mu^{\mathcal{P}^*_\theta}, \bar{\mu}_\theta^{\mathcal{P}^N_\theta}),$$

where "1" holds due to a substitution for equation 25, "2" holds due to triangle inequality.

$$\limsup_{N\to\infty} \mathcal{W}_2(\hat{\bar{\mu}}_T^{N,\mathcal{P}_\theta}, \mu^{\mathcal{P}^*_\theta}) + \limsup_{N\to\infty} \mathcal{W}_2(\mu^{\mathcal{P}^*_\theta}, \bar{\mu}_T^{\mathcal{P}^N_\theta}) = \limsup_{N\to\infty} \mathcal{W}_2(\mu^*, \bar{\mu}_T^{\mathcal{P}^N_\theta})$$

$$= \inf_{\mathcal{P}_\theta \in \mathcal{O}} \mathcal{W}_2(\mu^*, \bar{\mu}_T^{\mathcal{P}_\theta}), \tag{27}$$

Hence, $\mathcal{P}_\theta \mapsto \mathcal{W}_2(\hat{\mu}^N, \bar{\mu}_T^{\mathcal{P}_\theta})$ epi converges to $\mathcal{P}_\theta \mapsto \mathcal{W}_2(\mu^*, \bar{\mu}_T^{\mathcal{P}_\theta})$ $\mathbb{P}$-almost surely holds.

By the definition of epi convergence, Theorem 8.6, and Assumption 2.3. According to Assumption 2.2, we can find an $E$ satisfying that for a $\omega \in E$, where $\mathbb{P}(E) = 1$ Based on the above process, we can conclude that

$$\limsup_{N\to\infty} \inf_{\mathcal{P}_\theta \in \mathcal{P}_\Theta} \mathcal{W}_2(\hat{\mu}^N(\omega), \bar{\mu}_T^{\mathcal{P}_\theta}) \leq \inf_{\mathcal{P}_\theta \in \mathcal{P}_\Theta} \mathcal{W}_2(\mu^*, \bar{\mu}_T^{\mathcal{P}_\theta}) = \mathcal{P}_\theta^*. \tag{28}$$

By conditioning on the loss function and the definition of the infimum, there exists $N_1(\omega)$ such that for $\epsilon > 0$,

$$\inf_{\mathcal{P}_\theta \in \mathcal{P}_\Theta} \mathcal{W}_2(\hat{\mu}^N(\omega), \bar{\mu}_T^{\mathcal{P}_\theta}) \leq \inf_{\mathcal{P}_\theta \in \mathcal{P}_\Theta} \mathcal{W}_2(\hat{\mu}^n(\omega), \bar{\mu}_T^{\mathcal{P}_\theta}) + \mathcal{W}_2(\mu^n(\omega), \hat{\mu}^{N,\mathcal{P}_\theta}) \leq \mathcal{P}_\theta^* + \epsilon/2 \tag{29}$$

and the set

$$\{\mathcal{P}_\theta \in \mathcal{P}_\Theta : \mathcal{W}_2(\hat{\mu}^N(\omega), \bar{\mu}_T^{\mathcal{P}_\theta}) \leq \mathcal{P}_\theta^* + \epsilon/2\}, \tag{30}$$

is non-empty for all $n \geq N_1(\omega)$. By the triangle inequality,

$$\mathcal{W}_2(\mu^*, \bar{\mu}_T^{\mathcal{P}_\theta}) \leq \mathcal{W}_2(\mu^*, \hat{\mu}^N(\omega)) + \mathcal{W}_2(\hat{\mu}^N(\omega), \bar{\mu}_T^{\mathcal{P}_\theta}), \tag{31}$$

and 2.3, there exists $N_2(\omega)$ such that

$$\mathcal{W}_2(\mu^*, \hat{\mu}^N(\omega)) \leq \epsilon/2, \tag{32}$$

Then

$$\mathcal{W}_2(\mu^*, \bar{\mu}_T^{\mathcal{P}_\theta}) \leq \epsilon + \mathcal{P}_\theta^*. \tag{33}$$

This means that:

$$\{\mathcal{P}_\theta \in \mathcal{P}_\Theta : \mathcal{W}_2(\hat{\mu}^N(\omega), \bar{\mu}_T^{\mathcal{P}_\theta}) \leq \mathcal{P}_\theta^* + \epsilon/2\} \subset B(\epsilon). \tag{34}$$

Let $N_0 = \max(N_1(\omega), N_2(\omega))$ when $N \geq N_0$ we have

$$\inf_{\mathcal{P}_\theta \in \mathcal{P}_\Theta} \mathcal{W}_2(\hat{\mu}^N(\omega), \bar{\mu}_T^{\mathcal{P}_\theta}) = \inf_{\mathcal{P}_\theta \in B(\epsilon)} \mathcal{W}_2(\hat{\mu}^N(\omega), \bar{\mu}_T^{\mathcal{P}_\theta}). \tag{35}$$

According to 8.8, we can get the result:

$$\inf_{\mathcal{P}_\theta \in \mathcal{P}_\Theta} \mathcal{W}_2(\hat{\mu}^N(\omega), \bar{\mu}_T^{\mathcal{P}_\theta}) \to \inf_{\mathcal{P}_\theta \in \mathcal{P}_\Theta} \mathcal{W}_2(\mu^*, \bar{\mu}_T^{\mathcal{P}_\theta}),$$

and $N_0(\omega) = N$ the sets $\arg\min_{\mathcal{P}_\theta \in \mathcal{P}_\Theta} \mathcal{W}_2(\hat{\mu}^N(\omega), \bar{\mu}_T^{\mathcal{P}_\theta})$ are non-empty form a bounded sequence with

$$\lim_{N \to \infty} \sup \arg\min_{\mathcal{P}_\theta \in \mathcal{P}_\Theta} \mathcal{W}_2(\hat{\mu}^N(\omega), \bar{\mu}_T^{\mathcal{P}_\theta}) \subset \arg\min \mathcal{W}_2(\mu^*, \bar{\mu}_T^{\mathcal{P}_\theta}),$$

further we have:

$$\mathcal{P}_\theta^N \to \mathcal{P}_\theta^* \text{ as } N \to \infty, \epsilon_l \to 0, \mathbb{P}\text{-almost surely.} \tag{36}$$

$\square$

Next we give the proof of Theorem 2.8.

**Theorem 8.9.** *(Särkkä & Solin (2019)) If the SDE has strong solution 8.1, then the solution to the corresponding backward partial differential equation (PDE) can represent the expectation of the terminal distribution of the SDE.*

$$\frac{\partial u(x,t)}{\partial t} + \sum_{i=1}^d \frac{\partial u(x,t)}{\partial x_i} b_i(x,t) + \frac{1}{2} \sum_{i=1}^d \sum_{j=1}^d \frac{\partial^2 u(x,t)}{\partial x_i \partial x_j} (\sigma(x,t)\sigma(x,t)^T)_{i,j} = 0,$$

$$u(x,T) = f(x).$$

*The soluton of PDE at $(x_0, 0)$ is $\mathbb{E}[f(X_T)|X_0 = x_0]$, i.e. $u(x_0,0) = \mathbb{E}[f(X_T)|X_0 = x_0]$.*

*Proof.* According to Itô's formula

$$du(X_t,t) = \left[ \frac{\partial u(X_t,t)}{\partial t} + \sum_{i=1}^d \frac{\partial u(X_t,t)}{\partial x_i} b_i(X_t,t) + \frac{1}{2} \sum_{i=1}^d \sum_{j=1}^d \frac{\partial^2 u(X_t,t)}{\partial x_i \partial x_j} (\sigma(X_t,t)\sigma(X_t,t)^T)_{i,j} \right] dt$$

$$+ \left[ \sum_{r=1}^d \sum_{i=1}^d \frac{\partial u(X_t,t)}{\partial x_i} \sigma_{i,r}(X_t,t) \right] dW_t^r,$$

where $W_t^r$ is the rth component of $W_t$. The first part is based on the equality inside the PDE being set to zero. Integrate from $[0,T]$ on both sides.

$$u(X_T,T) - u(X_0,0) = f(X_T) - u(X_0,0) = \int_0^T \left[ \sum_{r=1}^d \sum_{i=1}^d \frac{\partial u(X_t,t)}{\partial x_i} \sigma_{i,r}(X_t,t) \right] dW_t^r,$$

Taking the conditional expectation on both sides while fixing $X_0$ and utilizing the properties of Itô integration as a martingale, we get

$$u(x_0,0) = \mathbb{E}[f(X_T)|X_0 = x_0].$$

$\square$

# 9 BACKGROUNDS AND TABLES OF THE EXPERIMENT

## 9.1 EXPERIMENTS ON THE ISING MODEL

Traditional MCMC algorithms face limitations when dealing with discrete random variables and complex functions $f$, which results in high variance. Consequently, accurate estimates often require a large number of points in the Markov chain, especially in larger models. This issue is particularly prevalent in random graph models (Cipra (1987); Newman et al. (2002); Drobyshevskiy & Turdakov (2019)).

**Ising model**: Assume a sample space $\Omega$, Hamiltonian function $H : \Omega \to \{0\} \cup [1, \infty)$, and inverse temperature parameter $\beta \in \mathbb{R}$, referred to as inverse temperature. The Gibbs distribution on $\Omega, H(\cdot)$, and $\beta$ is then characterized by probability law $\forall x \in \Omega : \pi_\beta(x) \doteq \frac{1}{Z(\beta)} \exp(-\beta H(x))$ Here $Z(\beta)$ is the normalizing constant or Gibbs partition function (GPF) of the distribution, with $Z(\beta) \doteq \sum_{x \in \Omega} \exp(-\beta H(x))$. Specifically, we considered Ising model on 2D lattices: It has $n \times n$ dimensions and a total of $n^2$ random variables, each of which takes the values +1,-1 with Hamiltonian function $H(x) = -\sum_{(i,j) \in E} \mathbb{I}(x(i) = x(j))$. For $\beta_0$ the results are easy to compute

and for $[\beta_1, \beta_2]$ between we can use the PPE-method, we do not use the Tpa-Method (Haddadan et al. (2021)), which is an algorithm on splitting the region $[\beta_1, \beta_2]$. Specifically we can compute the following $\mathbb{E}F = \mathbb{E}\exp(-\frac{\beta_2-\beta_1}{2}H(X_{\beta_1}))$ and $\mathbb{E}G = \mathbb{E}\exp(\frac{\beta_2-\beta_1}{2}H(X_{\beta_2})).X_{\beta_i}$ is a Gibbs distribution obeying parameter $\beta_i$. $Q = \frac{\mathbb{E}G}{\mathbb{E}F} = \frac{Z(\beta_1)}{Z(\beta_2)}$.We set $\beta_1 = -0.02$ and $\beta_2 = 0$. Then we can find $Z(\beta_1)$ based on the fact that $Z(\beta_2) = Z(0)$. So we need to estimate two mathematical expectations and we propose two ways to approximate this expectation. In this Experiment, for a definite temperature $\beta$, the distribution on the random graph is often easy to approximate, but the complexity of the exponent in the target expectation and also the function $H(x)$ can lead to the need for a large sample size to reduce the variance when MCMC deals with this problem. Our approach demonstrates superior efficiency in dealing with the distribution on a random graph, particularly when considering the complexities introduced by the target expectation exponent and the function $H(x)$. The implementation of our method, FKEE, stands out in handling larger-sized graphs ($n \geq 6$) where traditional MCMC and its variants, as found in (Haddadan et al. (2021)), face challenges due to sample complexity. We have the following two methods:

First method: direct approximation of the overall part of the expectation. That is, we consider the approximate stochastic process $H_\beta(X)$, which is a one-dimensional problem. We generate the chain using the same method as in (Haddadan et al. (2021)) and compute the value $H_\beta(X_t)$ under each moment. The diffusion bridge model and the Feynman-Kac model are then used to estimate the expectation separately. In the diffusion bridge model, we generated the same number of Brownian motions at the same number of moments and then calculated the loss at each moment to train. The Feynman-Kac model uses the already established diffusion bridge model to get an estimate of the expectation by solving the PDE.

The second approach better exemplifies the substantial improvement in harnessing Markov chains facilitated by our method. It highlights the remarkable flexibility embedded in our approach. Specifically, we directly approximate the distribution on a random graph, conceptualizing this graph as an $n^2$ random variable $(X_1, X_2, ..., X_{n^2})$, with each variable assuming two discrete values. A Markov chain is executed to obtain a sizable sample of random variables, and we subsequently approximate this $n^2$ dimensional distribution using a diffusion bridge model. However, since we are using a continuous model via SDE to obtain $Y_T$, which cannot accurately approximate a discrete random variable with values of $\{0, 1\}$, we employ the sigmoid function in the output $Y_T$. The loss for $X_T$ is then computed. Finally, when using the diffusion model, we apply post-processing to obtain the output value, i.e., $torch.round()$. In the case of the Feynman-Kac model, we set the boundary conditions to $u(x, T) = p(H(round(sigmoid(x))))$, where $p$ is $exp(-\beta/2*(x))$. In other words, we set the composite function $p(H(round(sigmoid)))$ to $f$ in the boundary.

Table 2 and Table 3 are one table. We have separated them for ease of presentation, and they have the same rows. In Table 2 and Table 3, $wi, vi, q$ represent the values of the corresponding $\mathbb{E}F, \mathbb{E}G, Q$ estimated using the corresponding estimators, respectively. $true\_wi, true\_vi, true\_z$ indicate the corresponding true values. The $error\_wi, error\_vi, error\_z$ represent the squared error using the corresponding estimators. The terms $w_i$ sample points and $v_i$ sample points refer to the number of sampled points utilized by the estimator. The terms $w_i$ time and $v_i$ time refer to the time taken by the estimator, measured in seconds. MCMC method we employed to generate samples follows the same approach as used in `https://github.com/zysophia/Doubly_Adaptive_MCMC`.

At the same time we compare with the method RelMeanEst in (Haddadan et al. (2021)). MCMC-C is the method RelMeanEst, MCMC-R is the empirical mean taken using the samples obtained from resampling, and MCMC-T is the estimate of the expectation obtained using the established diffusion bridge. And the number of data points used indicates the number of points in the Markov chain used. To be fair, we lower the threshold in MCMC-C to reduce its algorithmic complexity. Because only a small number of sample points are used in MCMC-R and MCMC-T. sample points means the number of points sampled from the Markov chain. Note that when $n \geq 6$ is in the MCMC-C method due to the larger complexity we do not discuss it. We only compare MCMC-R and MCMC-T. Note: GPU types: the first of these uses Tesla P100 while the second uses Tesla V100 when $n \geq 6$. The two methods are shown in Table 2 and Table 3. Above the horizontal is the first method below the second method. We can find the performance of PINN. In the high-dimensional case ($d = n^2 = 225$). The baseline model and hyperparameters in which the training was performed can be found in the code in our Supporting Materials.

Table 2: Comparison of different MCMC Expectation Estimator

| Method | $n$ | $wi$ | $vi$ | $q$ | $true\_wi$ | $true\_vi$ | $true\_q$ |
|--------|-----|------|------|-----|-----------|-----------|----------|
| MCMC-C | 2 | 0.9706396 | 1.0306606 | 1.0618365 | 0.9654024 | 1.0357122 | 1.072778 |
| MCMC-R | 2 | 0.9550395 | 1.0308957 | 1.0794273 | 0.9654024 | 1.0357122 | 1.072778 |
| MCMC-T | 2 | 0.9626546 | 1.0333116 | 1.073398 | 0.9654024 | 1.0357122 | 1.072778 |
| MCMC-C | 3 | 0.9340726 | 1.0744393 | 1.1502738 | 0.9226402 | 1.0834867 | 1.174333 |
| MCMC-R | 3 | 0.9269992 | 1.0774463 | 1.1622948 | 0.9226402 | 1.0834867 | 1.174333 |
| MCMC-T | 3 | 0.9283546 | 1.0795156 | 1.1628268 | 0.9226402 | 1.0834867 | 1.174333 |
| MCMC-C | 4 | 0.8844253 | 1.1470378 | 1.2969301 | 0.8641533 | 1.1563625 | 1.338233 |
| MCMC-R | 4 | 0.8686283 | 1.159192 | 1.3345087 | 0.8641533 | 1.1563625 | 1.338233 |
| MCMC-T | 4 | 0.8692993 | 1.1552249 | 1.328915 | 0.8641533 | 1.1563625 | 1.338233 |
| | | | | | | | |
| MCMC-R | 2 | 0.9950697 | 1.00498 | 1.0099593 | 0.9654024 | 1.0357122 | 1.072778 |
| MCMC-T | 2 | 0.9735975 | 1.0444663 | 1.0727907 | 0.9654024 | 1.0357122 | 1.072778 |
| MCMC-R | 3 | 0.9949918 | 1.0049603 | 1.0100187 | 0.9226402 | 1.0834867 | 1.174333 |
| MCMC-T | 3 | 0.9188372 | 1.0843412 | 1.1801233 | 0.9226402 | 1.0834867 | 1.174333 |
| MCMC-R | 4 | 0.9951742 | 1.0050679 | 1.0099418 | 0.8599499 | 1.1563472 | 1.344668 |
| MCMC-T | 4 | 0.8664092 | 1.1570783 | 1.3354871 | 0.8599499 | 1.1563472 | 1.344668 |
| MCMC-R | 6 | 0.9949221 | 1.0049597 | 1.0100888 | 0.7163408 | 1.3985122 | 1.9523 |
| MCMC-T | 6 | 0.6953082 | 1.3970394 | 2.0092378 | 0.7163408 | 1.3985122 | 1.9523 |
| MCMC-R | 8 | 0.9950855 | 1.0050602 | 1.010024 | 0.5468445 | 1.8348543 | 3.3553494 |
| MCMC-T | 8 | 0.5683886 | 1.9384431 | 3.4104189 | 0.5468445 | 1.8348543 | 3.3553494 |
| MCMC-R | 10 | 0.9949943 | 1.0050541 | 1.0101104 | 0.3853279 | 2.60382 | 6.7574135 |
| MCMC-T | 10 | 0.3684352 | 2.833073 | 7.6894751 | 0.3853279 | 2.60382 | 6.7574135 |
| MCMC-R | 15 | 0.9949888 | 1.0050285 | 1.0100903 | 0.1135434 | 8.894777 | 78.3381355 |
| MCMC-T | 15 | 0.1181741 | 10.4130456 | 88.1161667 | 0.1135434 | 8.894777 | 78.3381355 |

Table 3: Comparison of different MCMC Expectation Estimator

| $error\_wi$ | $error\_vi$ | $error\_q$ | $wi$ sample points | $vi$ sample points | $wi$ time (s) | $vi$ time (s) |
|-------------|-------------|------------|--------------------|--------------------|---------------|---------------|
| 2.74E-05 | 2.55E-05 | 0.000119716 | 3157 | 3157 | 0.29448 | 0.28554 |
| 0.00010739 | 2.32E-05 | 4.42E-05 | 100 | 100 | 14.421 | 9.702 |
| 7.55E-06 | 5.76E-06 | 3.84E-07 | 100 | 100 | 11.671 | 11.789 |
| 0.0001307 | 8.19E-05 | 0.000578845 | 30700 | 30700 | 3.40208 | 3.43238 |
| 1.90E-05 | 3.65E-05 | 0.000144918 | 2000 | 2000 | 20.163 | 19.544 |
| 3.27E-05 | 1.58E-05 | 0.000132393 | 2000 | 2000 | 236.109 | 233.225 |
| 0.000410954 | 8.70E-05 | 0.00170593 | 11383 | 11383 | 0.922 | 0.93 |
| 2.00E-05 | 8.01E-06 | 1.39E-05 | 2000 | 2000 | 26.917 | 26.696 |
| 2.65E-05 | 1.29E-06 | 8.68E-05 | 2000 | 2000 | 284.784 | 285.609 |
| | | | | | | |
| 0.000880149 | 0.000944468 | 0.003946189 | 500 | 500 | 7.562 | 5.728 |
| 6.72E-05 | 7.66E-05 | 1.61E-10 | 2000 | 2000 | 15.809 | 15.456 |
| 0.005234754 | 0.006166395 | 0.026999189 | 500 | 500 | 7.748 | 7.547 |
| 1.45E-05 | 7.30E-07 | 3.35E-05 | 2000 | 2000 | 32.482 | 32.562 |
| 0.018285611 | 0.022885427 | 0.112041629 | 500 | 500 | 10.762 | 10.573 |
| 4.17E-05 | 5.35E-07 | 8.43E-05 | 2000 | 2000 | 59.597 | 58.064 |
| 0.077607541 | 0.15488357 | 0.887761945 | 500 | 500 | 25.194 | 23.99 |
| 0.00044237 | 2.17E-06 | 0.003241913 | 2000 | 2000 | 302.916 | 303.36 |
| 0.200919994 | 0.688558248 | 5.500551232 | 500 | 500 | 64.449 | 62.8 |
| 0.000464148 | 0.010730639 | 0.00303265 | 2000 | 2000 | 516.126 | 509.603 |
| 0.371693119 | 2.556052403 | 33.03149292 | 500 | 500 | 91.826 | 90.796 |
| 0.000285363 | 0.052556938 | 0.868738826 | 2000 | 2000 | 889.684 | 906.444 |
| 0.776945993 | 62.24813139 | 5979.626574 | 500 | 500 | 165.086 | 166.38 |
| 2.14E-05 | 2.305139542 | 95.60989415 | 2000 | 2000 | 1968.976 | 1997.663 |

## 9.2 EFFECTS OF DIFFUSION BRIDGE MODEL

This method is also applicable for estimating integrals in high dimensions, particularly in tandem with the Monte Carlo method, when the target distribution is easily samplable. We conducted a com-

parison in estimating the distribution of the bridge using a bridge constructed from partially sampled high-quality samples. This approach enables continuous sampling of the target distribution by utilizing a well-established bridge. To illustrate, we simulated a diffusion bridge model (DBM) to approximate the distribution of a target variable $Y = (X_1, X_2, X_3)$, where $X_1 \sim N(1, 2) + Beta(4, 2)$, $X_2 \sim N(-1, 2) + Gamma(1, 2)$, and $X_3 \sim N(3, 2) + geometric(0.5)$. We sampled 500 points from the target distribution and employed DBM matching to obtain an SDE. Subsequently, we compared the distribution of the generated tracks to the target distribution. Continuing the target distribution sampling using the constructed bridge, we sampled an additional 500 points and compared the differences between the resampled samples and the original target distribution. The specific parameters include $T = 0.2$, time step size $h = 0.025$. The DBM is trained for 300 epochs with a learning rate of 0.001, using the Adam optimizer and Wasserstein distance as the loss. This method facilitates the construction of a pair of target distributions amenable to sampling. The expectation $\mathbb{E}(f(X))$ of the target distribution can be obtained by utilizing FCM.

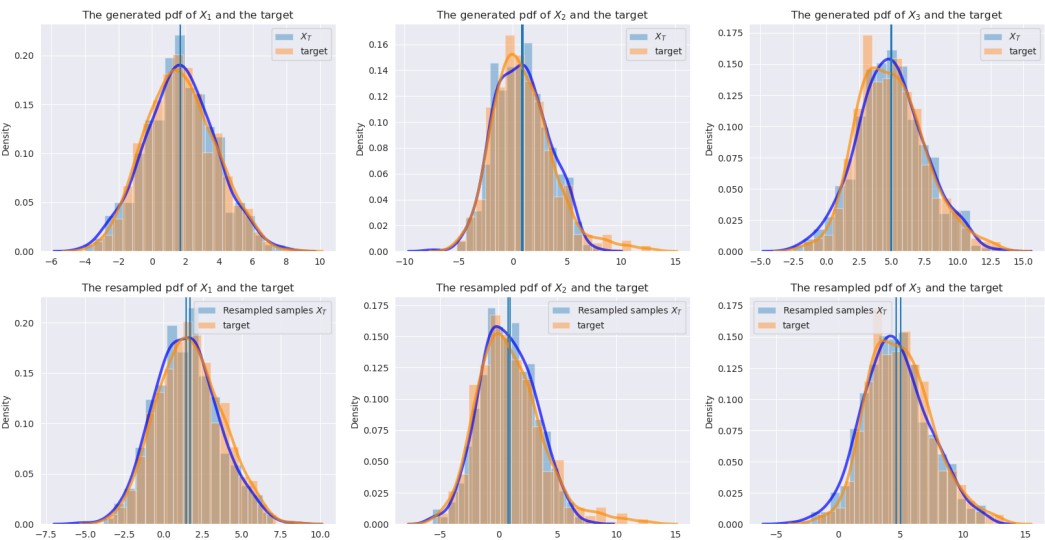

Figure 4: Comparison of the probability density functions of the generated and resampled paths and target distributions for each dimension. Two of the blue lines are the mean of the experience of the target sample and the mean of the experience of the re-generation sample, respectively.

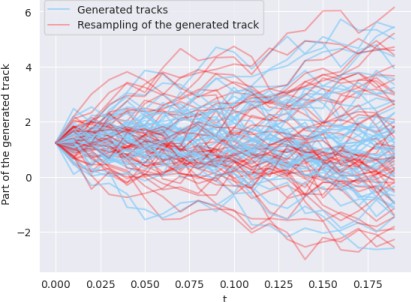

Figure 5: Generated tracks

### 9.3 OTHER BASELINE EXPERIMENTS

Since there are too many variants for the MCMC sampler, and our aim in this paper is to estimate the expectation rather than focusing on the selection aspect of the sampler, we consider one of the simplest LMCMC (Langevin diffusion model). It is worth noting, however, that we are using the

unadjusted LMCMC here.

$$dX_t = b(x)dt + dW_t,$$

where $b(x) = \frac{1}{2}\nabla_x \log p(x)$ and SDE solver is

$$X_{t+h} = X_t + b(X_t)h + W_{t+h} - W_t \quad X_0 = x_0$$

In all the experiments below, the interpretation of parameters is as follows: Total points in time: $D$, Initial value: $X_0 = x_0$, Brownian motion: $W_t$, Time Series: $t_0, t_1, \ldots, t_D = T$, Euler-Maruyama method step size: $h$, Number of paths simulated: $N$

We consider the MCMC algorithm and our method to sample the density function of the target and obtain the corresponding expectation. In the MCMC algorithm configuration, we use the Langevin MCMC to get independent samples. Here we use only the value of $X_T$ at the terminal moment to estimate the expectation. We use the same paths in LDM+FCM, but with a different way of computing expectations.

As an illustration, we consider a one-dimensional SDE, where we define the target distribution as $p(x) = C \exp(\frac{-(x-1)^2}{2})$, corresponding to the drift coefficient of the LDM being $\mu(x) = \frac{1-x}{2}$. We evaluate the expectation of $\mathbb{E}(X_{10})$. To decrease the error of the Euler-Maruyama method, we use a small step size of $h = 0.01$ and iterate 1000 steps to obtain the final path. We repeat the experiment $M = 30$ times. During the training process, we extract points from each path every 100 points and add them to the training process, instead of using all the points on the path.

We examine an extreme case, employing a very limited number of paths ($N = 5$) to estimate the true expectation $\mathbb{E}(X_{10})$. In Figure 1, we present the empirical distributions obtained through two different methods. The results obtained by LDM+FCM outperform Langevin MCMC, validating that paths can offer more informative outcomes. By incorporating gradient information from path points and integrating it into PINN for training, our method demonstrates lower variance under the same experimental configuration, significantly enhancing the efficiency of the MCMC algorithm with appropriate optimization. Although we utilize Unadjusted Langevin MCMC, our method provides unbiased estimates. **This is attributed to the fact that the bias in Unadjusted Langevin MCMC stems from the numerical SDE solver, while our method does not necessitate high accuracy in $X_t$; we are more concerned with the precision of the corresponding** $(b, \sigma)$ **on** $X_t$. Unlike direct sampling using the SDE method, which requires a highly precise SDE solver (Mou et al. (2021)), such precision is unnecessary in our method. We only require accurate estimations at each point on the path for the coefficients of the drift and diffusion terms.

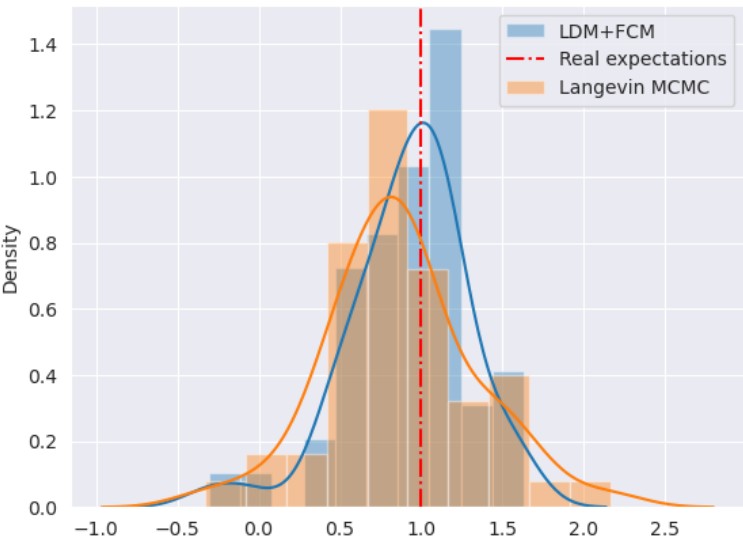

Figure 6: The empirical distribution of $\mathbb{E}_{estimated}(X_{10})$

For other cases, which can be handled by our method, we add an example of a broader computation of the expectation of a stable distribution in the absence of the corresponding convergence result for

MCMC. for example:

$$dX_t = \frac{1}{2}h^2 \frac{1 - 2X_t}{X_t^{\frac{1}{2}}(1 - X_t)^{\frac{1}{2}}}dt + 2hX_t^{\frac{1}{4}}(1 - X_t)^{\frac{1}{4}}dW_t$$

where $X_0 = 0.5, \quad \mathbb{E}X_1 = 0.5$

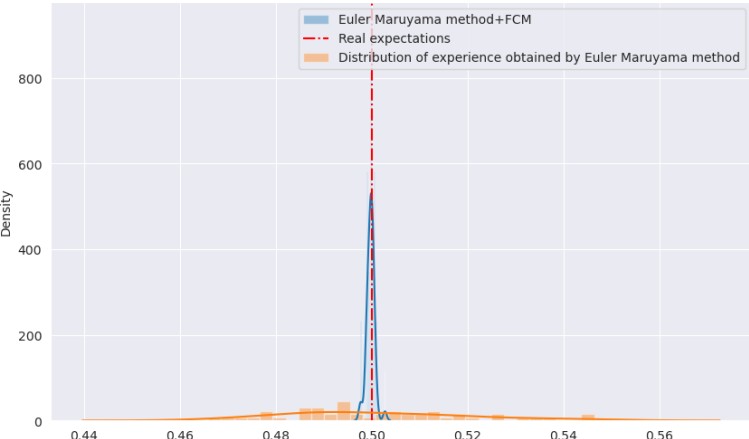

Figure 7: The empirical distribution of $\mathbb{E}_{estimated}(X_1)$

In this example, we use this method to estimate mathematical expectations in high dimensions. We consider a normal distribution with independent and identical marginal distributions as follows: $p(x) = C\exp(-0.5(x - 0.2)^2)$ for each dimension. When $g(x_1, \ldots, x_d) = x_1 + \cdots + x_d$, We use a smaller number of paths ($N = 50, 100$). We use the Euler-Maruyama method with a step size of 0. 1 for 100 iterations and calculate the internal loss function of the PDE every 10 points. In the case of $d = 5, 10$, we employ a 2-layer neural network with 108 units per layer and a *tanh* activation function. In the case of $d = 20$, we use a 2-layer neural network with 526 units per layer, set $N = 100$, and compute the internal loss function of the PDE every 20 points. We also repeate the experiment $M = 30$ times by using different random number seeds and measure the average time required to estimate the mathematical expectation each time with GPU (Tesla P100). In the training process we train 400 epochs by using the Adam optimizer with a learning rate of 0.001.

We compute $\mathbb{E}(g(X_1, X_2, \ldots, X_d))$ where $X_i \sim N(0.2, 1)$ and estimate its error. The errors we use is

$$\textbf{Absolute value error} = \frac{1}{M}\sum_{i=1}^{M}|\mathbb{E}_{estimated}^i(g(X_1, X_2, \ldots, X_d)) - \mathbb{E}(g(X_1, X_2, \ldots, X_d)))|$$

and

$$\textbf{Square Error} = \frac{1}{M}\sum_{i=1}^{M}|\mathbb{E}_{estimated}^i(g(X_1, X_2, \ldots, X_d)) - \mathbb{E}_{mean}(g(X_1, X_2, \ldots, X_d)))|^2$$

where

$$\mathbb{E}_{mean}(g(X_1, X_2, \ldots, X_d))) = \frac{1}{M}\sum_{i=1}^{M}\mathbb{E}_{estimated}^i(g(X_1, X_2, \ldots, X_d))$$

The method is LDM+FCM and we compare the results of this method with those obtained by the Langevin MCMC (LMCMC in short).

Table 4: Comparison of different methods

| Method | Dimension($d$) | paths($N$) | Absolute value error | Square Error | GPU time |
|---|---|---|---|---|---|
| LMCMC | 5 | 50 | 2. 927620e-01 | 1.253495e-01 | × |
| LDM+FCM | 5 | 50 | 1. 031084e-01 | 1.600998e-02 | 29. 62s |
| LMCMC | 10 | 50 | 4. 696985e-01 | 3.077161e-01 | × |
| LDM+FCM | 10 | 50 | 3. 330310e-01 | 1.382318e-01 | 46. 79s |
| LMCMC | 20 | 100 | 3. 630368e-01 | 1.863313e-01 | × |
| LDM+FCM | 20 | 100 | 2. 959023e-01 | 1.042063e-01 | 49. 74s |

## 9.4 POTENTIAL APPLICATIONS AND FUTURE WORK

**The independence of samples:** Obtaining accurate expectations with entirely unknown sample independence remains a significant challenge in the real world, particularly in stochastic optimization algorithms or loss functions, where independent sampling is frequently required for estimation. The independence of samples plays a crucial role in machine learning, and its violation can significantly impact the performance and validity of machine learning models. Many machine learning models rely on the assumption of independent and identically distributed (i.i.d) samples. Non-independent samples can introduce dependencies that the model may mistakenly learn as patterns. Nonlinear mathematical expectations play a critical role in such non-iid scenarios (Peng (2010)). However, methods like using Max-Mean Monte Carlo for calculating nonlinear mathematical expectations are often challenging. This is because we need to partition the dataset into parts where the samples are independent and then calculate the linear mathematical expectation for each part. Finally, we take the largest to get the nonlinear mathematical expectation. Our approach provides a completely new way to consider the use of Stochastic Differential Equations (SDEs) with G-Brownian motions. The diffusion bridge model is constructed using the same method and then solved directly using the Feynman-Kac model in the case of nonlinear mathematical expectations. This avoids problems such as data grouping.

**Representation learning and Distributional regression learning:** In the theory of statistical learning, we assume $X \sim P_X$ and $Y \sim P_Y$. A basic loss function is $l = \mathbb{E}[h_\theta(X) - Y)^2)]$ and $l = \mathbb{E}[\mathbf{Corss\ Entropy}[h_\theta(X), Y)]$ where $h_\theta$ is model, and we often need to sample a portion of the sample $\{x_i, y_i\}_{i=1}^N$, and then optimise the empirical loss function $l = \frac{1}{N} \sum_{i=1}^N (h_\theta(x_i) - y_i)^2$ and $l = \frac{1}{N} \sum_{i=1}^N \mathbf{Corss\ Entropy}(h_\theta(x_i), y_i)$. But in the case where the sample size does not fully cover the distribution of the corresponding totality, because the loss function is obtained by sampling a portion of the dataset, the loss function that we obtain tends to be biased, or has a large variance. When we have a high quality diffusion bridge that can accurately approximate the distribution of the target $(P_X, P_Y)$, which most of the current diffusion bridge models can do. We can achieve this by configuring the boundary conditions in the Feynman-Kac model to be $f(x, y) = (h_\theta(x) - y)^2$ and $\mathbf{Cross\ Entropy}(h_\theta(x), y)$. We then replace the empirical loss function with the PDE loss and the PDE loss at the boundary. This approach may enable us to enhance the learning of the **Representation of a Distribution**. This is because the diffusion bridge model captures information about the entire distribution rather than just the local distribution of specific points. When estimating expectations, we incorporate the PDE loss function, which contains gradient information regarding the diffusion bridge coefficients. The coefficients of the diffusion bridge tend to exhibit correlations with the target distribution. In this case, the number of points required for the diffusion bridge coefficients is often significantly smaller than the number of points $N$ directly sampled from the data. Finally, we can use the trained diffusion bridge model to perform some basic statistical learning tasks.

**Variational Inference:** Due to the extensive application of mathematical expectations in machine learning and probabilistic statistics, we are unable to comprehensively demonstrate all relevant methods in this paper. We will consider applying these methods to important domains, such as estimating the evidence lower bound (**ELBO**) in Black-Box Variational (Ranganath et al. (2014)) Inference. We often need to use **reparameterization** techniques to estimate the **ELBO** $= \mathbb{E}_{q(z|\phi)} \log p(x, z) - \log q(z|\phi)$ with small bias, but we can consider using a diffusion bridge to approximate the target distribution $q(z|\phi)$, and select $f(z) = \log p(x, z) - \log q(z|\phi)$, where $\phi$ can be designed as a trainable parameter. In this way, we can modify our optimization objective from

**ELBO** to $-u(x_0, t_0) + \texttt{PDE loss} + \texttt{boundary loss}$, which can achieve lower variance and GPU acceleration.

## 10 ALGORITHMS

For $X$ taking values in $\mathbb{R}^d$, if the marginal distribution is not independent, we employ one-dimensional Wasserstein distances (Santambrogio (2015)). In this case the Sinkhorn algorithm (Cuturi (2013)) can be used to address the optimal transport problem in $d$-dimensions.

**Backpropagation**: By encapsulating the computation of the 2-Wasserstein algorithm into an `nn.Module` and implementing it using `PyTorch`, we can retain the computation graph during the calculation. This allows us to obtain precise gradients using automatic differentiation methods.

---

**Algorithm 3** Diffusion bridge model (DBM)

---

**Input**: epochs:$M$,Total point in time:$D$,Learning Rate:$r$,Initial value:$X_0$,Brownian motion :$W_t$ Time Series: $t_0, t_1, \ldots, t_D = T$. Neural network: $b_{\theta_1}(x,t)$, $\sigma_{\theta_2}(x,t)$,$X_{0,\theta_3}$ and $\theta$ is the parameter of a neural network. Euler-Maruyama method of step $h$. Number of paths simulated $N$. $\varepsilon$ is the required error threshold. The given data point is $Y_T$.
**Output**: $X_i, b(t, X_i), \sigma(t, X_i), i \in [t_0, t_1, \ldots, t_D]$
1: Calculate $X_t$

$$X_{t+h} = X_t + b_{\theta_1}(t, X_t)h + \sigma_{\theta_2}(t, X_t)(W_{t+h} - W_t) \quad X_0 = X_{0,\theta_3}$$

2: **for** $k$ in $1 : M$ **do**
3:     Calculate loss

$$\mathcal{L} = \mathcal{W}_2(\hat{\bar{\mu}}_T, \hat{\mu}^N)$$

4:     **if** Match the whole Markov chain **then**
5:         Calculating the loss of this path assumes a Markov chain with $M$ steps

$$\mathcal{L} = \sum_{i=1}^{M} \mathcal{W}_2(\hat{\bar{\mu}}_{t_i}, \hat{\mu}^*_{t_i})$$

        .
6:     **end if**
7:     **for** $n$ in $1 : 3$ **do**
8:         Update parameters $\theta_n^k \leftarrow \theta_n^{k-1} - \nabla_\theta \mathcal{L} r$.
9:     **end for**
10:    **if** $\mathcal{L} < \varepsilon$ **then**
11:        End of training.
12:    **end if**
13: **end for**

---

The algorithm of the Feynman-Kac model is similar to that of PINN. It is mainly a matter of using the diffusion coefficients obtained earlier and solving the corresponding PDE for the data points. Not all points on the paths need to be included in the training in this algorithm. This is the same as the training of PINN, where we only need to sample a fraction of the points to get the solution.

---

**Algorithm 4** Feynman-Kac model (FCM)

---

**Input**: epochs:$M$ ,Total point in time:$D$ ,Learning Rate: $r$ ,Time Series: $t_0, t_1, \ldots, t_D = T$. Points of observation :$X_t$,Drift coefficient: $b(t, X_t)$,Diffusion coefficient:$\sigma(t, X_t)$ where $t \in [t_0, t_1, \ldots, t_D]$Neural network: $u_\theta(x, t)$ $\theta$ is the parameter of a neural network. The function $f$ that needs to be estimated. Number of paths simulated $N$. required error threshold $\varepsilon$.

**Output**: $\mathbb{E}(f(X_T)|X_0 = x_{t_0}) = u_\theta(x_{t_0}, t_0)$

1: **if** $\sigma(t, X_t)$ is the diagonal matrix **then**
2:    **for** $k$ in $1 : M$ **do**
3:       **for** $s$ in $1 : D - 1$ **do**
4:

$$\mathcal{L}_1^s = \frac{1}{N} \sum_{k=1}^N \left\{ \frac{\partial u_\theta(x,t)}{\partial t} + \sum_{i=1}^d \frac{\partial u_\theta(x,t)}{\partial x_i} b_i(x_t, t) + \frac{1}{2} \sum_{i=1}^d \frac{\partial^2 u_\theta(x,t)}{\partial x_i^2} \sigma(x,t)_i^2 \bigg|_{(x,t)=(x_s^k, t_s)} \right\}^2$$

5:       **end for**
6:    **end for**
7: **end if**
8: **if** $\sigma(t, X_t) \neq$ diagonal matrix **then**
9:    **for** $k$ in $1 : M$ **do**
10:       **for** $s$ in $1 : D - 1$ **do**
11:

$$\mathcal{L}_1^s = \frac{1}{N} \sum_{k=1}^N \left\{ \frac{\partial u_\theta(x,t)}{\partial t} + \sum_{i=1}^d \frac{\partial u_\theta(x,t)}{\partial x_i} b_i(x, t) + \frac{1}{2} \sum_{i=1}^d \sum_{j=1}^d \frac{\partial^2 u_\theta(x,t)}{\partial x_i \partial x_j} (\sigma(x,t)\sigma(x,t)^T)_{i,j} \bigg|_{(x,t)=(x_s^k, t_s)} \right\}^2$$

.
12:       **end for**
13:    **end for**
14: **end if**
15: Calculate PDE loss.

$$\mathcal{L}_1 = \sum_{s=1}^{D-1} \mathcal{L}_1^s$$

.
16: Calculate boundary loss

$$\mathcal{L}_2 = \frac{1}{N} \sum_{k=1}^N \left\{ u_\theta(x_{t_D}^k, t_D) - f(x_{t_D}^k) \right\}^2$$

17: Update parameters $\theta^k \leftarrow \theta^{k-1} - \nabla_\theta(\mathcal{L}_1 + \mathcal{L}_2)r$
18: **if** $(\lambda_1 \mathcal{L}_1 + \lambda_2 \mathcal{L}_2) < \varepsilon$ **then**
19:    End of training
20: **end if**

---