# OpenReview forum: "Feynman-Kac Operator Expectation Estimator"
_ICLR.cc/2025/Conference — Submitted to ICLR 2025_

### Official Review · Reviewer_TUPv · 2024-11-03

**Soundness:** 3
**Presentation:** 2
**Contribution:** 2
**Rating:** 3
**Confidence:** 3

**Summary:**

The manuscript combines several DNN-based numerical techniques, associated with solving and learning SDEs and their PDE counterparts, with the aim of sampling from distributions and averaging over observables. Specifically, a type of neural-SDE is put forward and trained to converge to the desired distribution. Next averages of functions are estimated using the Feynman-Kac formula, wherein a PDE is derived from the learned SDE which “calculates” the averages for us given that one sets the boundary condition to be the function one wishes to average. This complex problem is solved using a PINN approach. Some numerical results on random graphs are reported in a table.

While the paper suggests an interesting technical path to explore in this important subdomain MC samplers— it is not clear from reading the manuscript how much of this is incremental work stringing together a series of known results or rather an original and meaningful step forward. While I am only a causal user of Monte-Carlo, my strong feeling is that the benchmarks shown are insufficient to prove that
this specific combination of techniques outperforms existing ones (including DNNs/PINN-based ones and more advanced MC techniques). Turning to the techniques themselves, it seems that neural-SDEs preceded the current work [Tzen, Ragidsky 2019], and that using Feynman-Kac formula with a combination of DNNs to estimate observables on an SDE has also been done before [Blechshmidt, Ernst 2021 and refs therein]. Reading into the latter work, it seems that DNNs have been used slightly differently than in the current work to solve the PDE associated with the SDE, but is this a conceptual change given the common use PINNs to solve PDE? Does it hold the key to any SOTA results? The related work section (which has been delegated to the appendix) and the general causal referencing to related works (such as Tzen et. al.) leave the non-expert reader with little understanding of the true novelty of the current results.

The presentation of the work also leaves a gap between conceptual claims and practical contributions. It also feels fragmented and the
common use of signposts and bold notation only worsens this in my mind. The conceptual claims, which are sometimes grand, are hard to substantiate. For instance, “Establishing a Link Between Sampling Methods and High-Dimensional Partial Differential Equations”, taken at face value, can hardly be attributed to the current work with all the knowledge on SDE, Fokker-Plank Equations, and Feynman-Kac formula. Also “Expanding the Scope of Expectation Estimators“, feels vague. Is there a current scope of expectation estimators? What results in the current work expand this scope in a way that others can't? Finally, in their introduction, the authors allude to the fact that the authors have an affirmative answer to the question “Is it possible to unify most existing MCMC algorithms into a cohesive framework to create a universal sampler for expectation estimation?“--- This is such a rich and complex problem that providing an affirmative answer would clearly violate various prevalent complexity theory assumptions. For instance, can the authors show that their sampler solves the Ising Spin-Glass problem? Can the authors even solve the much simpler case of the 2d Ising model at the phase transition and compute long-range observables? Does their technique outperform various ones used in physics to overcome sampling problems such as replica-exchange Monte Carlo?

The above issues, concerning the entanglement with previous works, evidence of going beyond SOTA, and its portrayed grand scope, prevent me from recommending it for publication in ICLR.

**Strengths:**

The work addresses an important fundamental topic. It provides an interesting combination of DNN-based techniques.

**Weaknesses:**

Relation to past literature is left vague. It does not show SOTA on a sufficient set of canonical problems with a suitable comparison to
other techniques. Its claims feel too broad and general. The writing style can be improved.

**Questions:**

Can the author disentangle their works from past literature on neural SDE and Feynman-Kac's use of averaging observables?

Can the authors provide evidence that their universal sampler outperforms the existing techniques, including those in Blechshmidt
et. al.(2021)? Can they provide some canonical well-excepted benchmark at which they excel over others?

---

> ### Author Response · Authors · 2024-11-13
> **Responses**
>
> Thank you for reading our paper in detail! Below are my responses to some of your statements. If any of them are not precise enough, feel free to ask more questions.
>
> The main contribution of this paper is a different interpretation of the FE formula, which can enhance the performance of most diffusion models (samplers) in estimating expectations. Blechschmidt and Ernst (2021) is a classic method for solving the FK equation, meaning that we can solve high-dimensional PDEs by simulating SDEs (which typically do not have dimensionality issues) and then computing expectations. This is just one way of utilizing the PDE. We shift this approach by assuming that we know the SDE is a diffusion (diffusion bridge) model, and accordingly, we can naturally express the corresponding PDE. By using a different method to solve the PDE, namely PINN, we can obtain the expectation of the SDE at the terminal point, which corresponds to the solution at the initial time of the PDE.
>
> This method avoids the need for LLN and ETMC to compute the mathematical expectation, and we often only need a small number of points to achieve better results. This is because the PDE integrates all the information of the SDE, and the SDE contains information such as the density, for example, in Langevin dynamics and score functions in score-based SDEs.
>
> Therefore, our validation experiments are centered around expectation estimation, rather than the sampler(MCMC) and PDE.
>
> Expanding the Scope of Expectation Estimators can be specifically referenced in line 415. In other words, we can integrate a large number of SDE samplers to expand the range of  $P$ , which may not even be a stationary distribution in MCMC. This weakens the conditions on $f $, as a larger Lipschitz constant for $f$ often leads to larger variance.
>
> Integrating MCMC into a unified framework has been recently achieved by many diffusion models, including our method, as seen in [1][2][3]. This represents the latest direction in current technology. However, there is a lack of corresponding work on efficiently obtaining expectations from SDEs, as most estimators still rely on ETMC and LLN.
>
> [1] Vargas, Francisco, Will Sussman Grathwohl, and Arnaud Doucet. "Denoising Diffusion Samplers." The Eleventh International Conference on Learning Representations.
>
> [2]Richter, Lorenz, and Julius Berner. "Improved sampling via learned diffusions." The Twelfth International Conference on Learning Representations.
>
> [3]Grenioux, Louis, et al. "Stochastic Localization via Iterative Posterior Sampling." Forty-first International Conference on Machine Learning.
>
> Our experiments demonstrate that we have obtained more accurate estimates of the partition function with fewer samples. Sampling algorithms and expectation estimators are fundamentally different methods. As we have emphasized, our focus is not on the MCMC method itself, but rather on whether MCMC + FE outperforms traditional methods. Our experiments have already shown that this is the case. Replica-exchange Monte Carlo is merely a sampling method and does not provide an expectation estimator. Moreover, the estimator will still be influenced by the conditions on $ f $.
>
> Questions
>
> Can the author disentangle their works from past literature on neural SDE and Feynman-Kac's use of averaging observables?
>
> Can the authors provide evidence that their universal sampler outperforms the existing techniques, including those in Blechshmidt et. al.(2021)? Can they provide some canonical well-excepted benchmark at which they excel over others?
>
> Answers see above.We welcome and accept any suggestions for improving the expression and clarity of the paper.

---

> ### Comment · Reviewer_TUPv · 2024-11-27
>
> I thank the authors for their response. I still maintain that the manuscript suffers, above all, from various presentation flaws which make it difficult to understand the true positioning of this work within the existing literature and assess its novelty. The response of the other referees only strengthened my view. I’d advise the authors to focus on the classical form of an introduction: explaining the open problem, providing motivation, reviewing what has been done, fleshing out a smaller open problem, and then explaining how their approach made steps towards solving this smaller one. Moreover do not assume that the reader understands the specific challenge you are facing.
>
> For instance, the current introduction poses the question "can we have a universal MCMC sampler” (paraphrasing question A). This is a far too general scope that is not even used as a "funnel” to dive into the specific problem the authors try to solve. Having some experience with MCMC methods, in many relevant settings getting uncorrelated samples is an NP-hard or sharp-P problem (e.g. spin-glass problems or their SAT-3 versions in computer science). Clearly one has to rely on the specifics of the problem at hand to make progress.
>
> Following this I still do not understand what is the novelty of the current work. As far as I understand, the authors are not the first to use the FK formula for expectation estimation. If their novelty is in an “interpretation”, it is too vague and subjective to my taste. If their technical innovations led to some obvious speedup, then the benchmarks given are still insufficient in my mind. In the physics community, the 2d Ising model was studied up to number of sites (n) of the order of millions. Furthermore, this was done at so-called critical points where simple local Monte-Carlo updates, reminiscent of their SDE, suffer from critical slowing down (which was then cleverly avoided using Worm Algorithms and Cluster Updates). Nonetheless, exact theoretical results are available for these large-scale systems and showed agreement with numerics up to 5-6 digit accuracy. In contrast, the authors report experiments on n=15^2 sites and do not tune the system to a critical point. It may be that the author actually put some different emphasis on which they do excel, but then this comes back to issues of presentation.
>
> If on the other hand, it is that they are more efficient per sample, then similar issues can be raised. First, as before, why is the benchmark carried on such small systems? How do we know this would scale to truly large-scale systems? Second, how much do these results depend on the samplers one is using? What if you used cluster updates that have very short burn-in periods accompanied by standard expectation estimation? Does the time complexity involve training the PINN and learning the SDE? How would that scale and perform when the dimension of the problem grows even larger?
>
> To summarize, I strongly believe the authors need to revise their presentation, delve into the specifics of their contribution, and explain how their numerical benchmark supports their claim. I also believe that the discussion period is not the time to make such an overhaul.

---

### Official Review · Reviewer_UFuF · 2024-11-03

**Soundness:** 2
**Presentation:** 2
**Contribution:** 2
**Rating:** 1
**Confidence:** 5

**Summary:**

The paper presents two generative models to use in the context of sampling: a diffusion bridge with $W^2$-loss and algorithm based on solving the Feynmann-Kac PDE using PINNS.

**Strengths:**

Developing new sampling methods using ideas from generative model is an active area with lots of promising recent advances.

**Weaknesses:**

+ There is no really new theoretical contributions in this paper. All the theorems mentioned in the paper are standard results. The diffusion bridge model has been used in various iterations in countless papers in the literature, for example by Doucet and collaborators.

+ There is no indication that the algorithms presented here will scale up with dimension, especially if using the W^2 loss, so the claim that this improves on MCMC seem somewhat overblown.

+ Using a PINNs to solve the Feynmann-Kac equation is very unlikely to work in high dimension as PINNs are usually are no easy to to train.

+ Lack of experiments on high-dimensional data sets.

+ No head-to-head comparisons with state of the art algorithms.

**Questions:**

+ What are the limitations  of your methods regarding dimension?

+ How does your algorithm compare with other neural ODE/SDE, diffusion models, bridges models in the literature?

---

> ### Author Response · Authors · 2024-11-13
> **Responses**
>
> What we propose in this paper is not a sampler or a generative model, but rather a heuristic algorithm that can enhance the performance of most existing SDE-based samplers in estimating expectations.
>
> Theoretically, we have provided the convergence results for our diffusion bridge model. The diffusion bridge model is merely a sampler designed to reduce the total training effort of the PINN. We reverse-engineer the FK equation, which is itself a conceptual innovation. Most of our past work focused on solving high-dimensional PDEs, namely using MCMC / SDEs and then compute the expectation to obtain the solution of the PDE. Our approach, however, is to directly solve the PDE to obtain the MCMC expectation. When the SDE is a diffusion model-based sampler, we can obtain the expectation of the target distribution without relying on the LLN or ETMC.
>
> Samplers using the $W_2$ distance are quite common, such as those based on optimal transport methods. For high-dimensional cases, Since the dimension is hidden in our Theorem 2.7, a classical convergence result is given in [1].
>
> [1] Fournier, Nicolas, and Arnaud Guillin. "On the rate of convergence in Wasserstein distance of the empirical measure." Probability theory and related fields 162.3 (2015): 707-738.
>
> We compare the SOTA expectation estimators and demonstrate the performance of PINN in the high-dimensional case with $d = 225$, as shown in the experiment in line 1029.
>
> How does your algorithm compare with other neural ODE/SDE, diffusion models, bridges models in the literature?
>
> Our goal is to estimate the expectation, not to sample. Therefore, our experiments focus more on the combination of the sampler + FE and sampler + ETMC/LLN, with the choice of sampler being diverse.

---

### Official Review · Reviewer_A2pH · 2024-11-04

**Soundness:** 2
**Presentation:** 2
**Contribution:** 2
**Rating:** 5
**Confidence:** 4

**Summary:**

The authors proposed to leverage the Feynmann-Kac equation via Physically Informed Neural Networks (PINN) to approximate the target Mathematical Expectation efficiently and heuristically without causing a large variance.

**Strengths:**

the idea of using Feyman-Kac to approximate the expectation is interesting.

**Weaknesses:**

The scalability of the algorithm w.r.t. dimension is not verified sufficiently. d=20 is too small. There are no real-world simulations.

The authors criticize the large variance issue by the MCMC method but fail to justify theoretically why the proposed method yields a lower variance. The empirical support is limited.

NIT: Theorem 2.1: the discretization error by Growall inequality is weak and exponentially dependent on time. Girsanov can be used to fix it.

**Questions:**

1. I don't know why and when MCMC is required to impose complex constraints on the distribution and performance function. Some references are suggested.

2. I don't see when MCMC is not the optimal decoding method. It appears to me that burn-in is not a significant limitation and only affects the performance on a negligible scale and can be easily fixed via a large stepsize in the beginning for warm-up.

3. Discussions on the limitations would be preferred.

---

> ### Author Response · Authors · 2024-11-13
> **Responses**
>
> Q1 The scalability of the algorithm w.r.t. dimension is not verified sufficiently. d=20 is too small. There are no real-world simulations.
>
> A1 In fact, we demonstrated the effectiveness of this method in high-dimensional cases, such as 225 dimensions, in the ising model. A detailed explanation of this experiment can be found in line 1077.
>
> Q2 The authors criticize the large variance issue by the MCMC method but fail to justify theoretically why the proposed method yields a lower variance. The empirical support is limited.
>
> A2 Theoretically, concentration inequalities can show that a larger Lipschitz constant will lead to a larger variance, which can be referenced in [1]. On the application level, the precision of the sampler also affects the variance. For example, using the Metropolis Adjusted Langevin Algorithm for correction. Our two experiments validate these two cases. First, for the ISING model case, the complexity of $f$ is the main factor. In the second case, we use the unadjusted Langevin Algorithm to obtain better results, without changing the structure of the sampler.
>
> The metric used in Girsanov's theorem is the KL divergence, but KL divergence does not have the properties of $W_2$. Currently, we are more focused on $W_2$. Additionally, Girsanov's theorem cannot be applied when the diffusion coefficients are inconsistent.
>
>
> [1] Gobet, Emmanuel. Monte-Carlo methods and stochastic processes: from linear to non-linear. Chapman and Hall/CRC, 2016.
>
> Q4 I don't know why and when MCMC is required to impose complex constraints on the distribution and performance function. Some references are suggested.
>
> A4 In fact, the example in the ISING model illustrates this point. In a more general case, when we want to solve an energy model such as $p(x;\theta) = \frac{\exp(-U(x,\theta))}{Z(\theta)}$, the partition function $Z(\theta)$ is intractable, especially when the state space of $x$ is discrete. This type of model is commonly found in large language models. Below are examples of handling $Z(\theta)$. In addition, in statistical inference, for example, when estimating the parameters of random graph models, this issue arises. While we can use MCMC sampling distributions, we cannot obtain an exact expression for the distribution due to the presence of the partition function.
>
>
> [2] Xu, Minkai, et al. "Energy-Based Diffusion Language Models for Text Generation." arXiv preprint arXiv:2410.21357 (2024).
>
> [3] Rafailov, Rafael, et al. "Direct preference optimization: Your language model is secretly a reward model." Advances in Neural Information Processing Systems 36 (2024).
>
>
> Q5 I don't see when MCMC is not the optimal decoding method. It appears to me that burn-in is not a significant limitation and only affects the performance on a negligible scale and can be easily fixed via a large stepsize in the beginning for warm-up.
>
> A5 The reason for this is quite simple: in diffusion models, $X_T$ does not necessarily have to be a stationary distribution. For example, methods like normalized flow methods and many diffusion model-based samplers can all use FE to solve for expectations without relying on ETMC and LLN. For MCMC objectives, diffusion bridge modelling is a more efficient approach [1].
>
>
> [1] Vargas, Francisco, Will Sussman Grathwohl, and Arnaud Doucet. "Denoising Diffusion Samplers." The Eleventh International Conference on Learning Representations.
>
> We appreciate the suggestion to discuss the limitations.

---

### Official Review · Reviewer_Su4v · 2024-11-04

**Soundness:** 2
**Presentation:** 2
**Contribution:** 2
**Rating:** 3
**Confidence:** 3

**Summary:**

The authors propose the Feynman-Kac Operator Expectation Estimator (FKEE) to approximate the target distribution E[f(X)]. This estimator contains two parts: (1) A diffusion bridge model with parameters optimized to minimize the Wasserstein distance to the target distribution, and (2) a method based on the Feynman–Kac equation, formulated as a partial differential equation (PDE) and solved approximately using Physics-Informed Neural Networks (PINNs), which employ a least-squares approach. The experiments focus on approximating the partition function in a random graph model.

**Strengths:**

A significant strength of this work is the innovative linking of the diffusion model to high-dimensional partial differential equations (PDEs), with Physics-Informed Neural Networks (PINNs) effectively employed to overcome the curse of dimensionality in solving these PDEs.

**Weaknesses:**

1. The Feynman–Kac model (Algorithm 2) with the PINN solver lacks a convergence or error estimate, which would be valuable for assessing its accuracy and reliability.

2. In the experiments, the authors claim that "using fewer points on the Markov chain achieves higher accuracy in approximating expectations." However, it is unclear if this result generalizes beyond the specific example provided, as it appears quite context-dependent.

**Questions:**

The authors mention in the Discussion that their method requires the boundary conditions of the PDE to satisfy a smoothness condition, specifically that f is in C^2, and that this requirement broadens the scope of their approach. However, it seems that C^2 smoothness could be more restrictive than a Lipschitz assumption. Could the authors clarify how they view this requirement as less restrictive? Additionally, could they discuss any potential limitations this might introduce for functions that are not in C^2?

---

> ### Author Response · Authors · 2024-11-13
> **Responses**
>
> Q1 The Feynman–Kac model (Algorithm 2) with the PINN solver lacks a convergence or error estimate, which would be valuable for assessing its accuracy and reliability.
> A1 Yes, this is indeed a problem. We provide some empirical results, but theoretically, we have actually cited the relevant convergence results [1] for such PDEs in line 380.
>
> [1]Tim De Ryck and Siddhartha Mishra. Error analysis for physics-informed neural networks (pinns)
> approximating kolmogorov pdes. Advances in Computational Mathematics, 48(6):79, 2022.
>
> Q2 In the experiments, the authors claim that "using fewer points on the Markov chain achieves higher accuracy in approximating expectations." However, it is unclear if this result generalizes beyond the specific example provided, as it appears quite context-dependent.
>
> A2  We mainly emphasize the statement in line 485. The main text only provides a rough result, and more details about this experiment can be found in the Appendix 9.1.
>
> Questions about conditions on functions
>
> A The function condition we are referring to is in the context of expectation estimators. In this field, Lipschitz functions will significantly affect the estimator's equation. For more details, you can refer to Section 2.4.4 in [1] and other concentration inequalities. Our assumption only requires the function to belong to $C^2$ and does not impose any restrictions on the Lipschitz constant.
>
> Potential limitations this might introduce for functions that are not in C^2, this could lead to potential issues in the convergence of PINN training, such as the generation of large gradients at the boundaries for functions that are not in $C^2$.
>
>
> [1] Gobet, Emmanuel. Monte-Carlo methods and stochastic processes: from linear to non-linear. Chapman and Hall/CRC, 2016.

---

> > ### Comment · Reviewer_Su4v · 2024-11-27
> >
> > Q1:  The authors should explicitly explain how these theoretical results in [1] apply to their method to strengthen the paper's clarity.
> >
> > The assumption that the function belongs to a certain space without constraining the Lipschitz constant is noted, but the justification for replacing this constraint with higher regularity is unclear. This seems more like a trade-off than a clear improvement. Further clarification or justification would strengthen the argument, especially regarding the impact on expectation estimators and the relevant equations.

---

> > > ### Author Response · Authors · 2024-11-27
> > > **Responses**
> > >
> > > First, the neural network structure we used is consistent with that in [1], employing the tanh activation function. In our setup, the training points (t,xt) and the loss function are also aligned with those in [1]. Theorem 3.1 in [1] presents the approximation theorem for neural networks, which serves as an existence theorem. Theorem 3.3 in [1] provides information about the parameters of the neural network. Finally, Theorem 3.7 establishes that the error can be controlled by the loss function. We include these results in the appendix as a concise form of support.
> > >
> > > This perspective focuses on addressing the problem within a specific domain. Taking our Ising model example, it is easy to demonstrate that the function $ \exp(x) $ satisfies $ C^2 $ smoothness but is not Lipschitz continuous. Using classical estimators in such cases often leads to significant bias. Another illustrative example is the classical Monte Carlo integration case, where $ f = x^n $. When $ n $ becomes sufficiently large, any estimator will introduce bias.
> > >
> > > Functions satisfying $ C^2 $ smoothness are more common and natural in the field of statistical probability, such as loss functions in Bayesian analysis/(ML) or energy-based models. For functions with large Lipschitz constants, classical estimation errors can often be bounded using concentration inequalities. However, for $ C^2 $ functions, there are currently no corresponding concentration inequalities without imposing bounds on the $ C^2 $ norm. This inherently expands the scope of applicable statistical models.

---

### Official Review · Reviewer_xwQ7 · 2024-11-04

**Soundness:** 1
**Presentation:** 1
**Contribution:** 1
**Rating:** 3
**Confidence:** 4

**Summary:**

Doing MCMC is hard (time consuming, and somewhat wasteful because of the burn-in period). The authors propose a post-processing method using the samples from some MCMC procedures, which admit an Itô decomposition, to approximate moment estimates of the desired sampling density. They use a denoising technique based on physics informed neural networks as part of the post-processing mechanism.

**Strengths:**

The goals stated in the introduction are bold, and quite interesting. Obtaining results in this line would prove quite useful in general for ML and statistics.

I appreciate that the authors include small introductions for the Euler-Maruyama method and physics informed neural networks in the appendix. However their existence should be indicated in the main text.

**Weaknesses:**

Presentation is bad throughout. There are plenty of typos. The authors do not use parenthetical citations and instead insert them in the text which makes for a less pleasant reading experience.

The notation introduced in line 232 definitely needs improvement, I do not understand which side is supposed to be the one that will be used later. Even then, it is unclear what is being defined, as there are two definitions for $\hat{\mu}\_{t\_i}$ .

In Assumption 2.2, it is not clear what $\mu^{\mathcal{P}_\theta}$ means.

The notation in Algorithms 1 and 3 should be introduced before the algorithms. For example, it is not clear where $Y_T$ comes from, and why it is required.

Table 1 is impossible for me to parse. I invite the authors to mimic the conciseness of their own Table 5 for summarizing the numerical results.

There is a link to GitHub page, which has been confirmed to not belong to the authors, but I do not see a good reason for why the authors would want to include a link to a GitHub that is not theirs. Usually a citation to the original article is enough.

The proofs in the main text for Theorems 2.1, 2.6 and 2.7 should either refer directly to the Appendix where they are proved, or be proved right there.

Regarding Theorem 2.8, the comment in the 'proof' space makes me think the Authors were not the first to prove it, in which case they should indicate it explicitly; otherwise it would be plagiarizing.

Currently Section 4 is quite lacking, including the aforementioned Table 1 which I cannot comprehend (by the way, it is missing a reasonable caption).

A proposed method like this should be thoroughly tested, which in the current state of the paper it has not been. The methods the authors refer for comparison should include appropriate references.

**Questions:**

In Theorem 2.1, what does "Linear growth" mean?

In Assumption 2.4, what does "D is the metric of the parameter" mean?

In Theorem 2.6, should "we exist" be "there exists a set"?

In Algorithm 1, can the authors clarify what is the main difference between $X_t$ and $X_i$? The distinction is not clear to me.

How are the integrals in Algorithm 2 computed? Are the authors able to evaluate the integrals explicitly? If so, they should indicate how and why they are able to do so.

How does the computational cost of this approach compare to other MCMC approaches?

One of the main criticisms posed about MCMC methods is that they are not optimal since they spend quite some time in the burn-in phase (lines 56, 82). However, for the proposed method to work the authors assume that they have access to samples from a distribution that are obtained via an MCMC, that has already gone through a burn-in period (lines 188, 192). How can the authors support their claim that this method is better (line 107) than MCMC if it is still spending a similar amount of samples in burn-in?

---

> ### Author Response · Authors · 2024-11-13
> **Responses**
>
> Thank you very much for your careful reading and for pointing out the unclear expressions in the paper. We have made the necessary corrections in a revised version. It has now been updated.
>
> Here are some responses to the questions:
> Q1 In Theorem 2.1, what does "Linear growth" mean?
> A1 On line 815, we provide the definition in the appendix.
>
> Q2 In Assumption 2.4, what does "D is the metric of the parameter" mean?
>
> A2 $P$ is a parameter of a triplet. We define the metric, which can be considered in the product space, for example, the first for $X_0$ is the Euclidean 2-norm of vectors, the second for $b$ is the $L_2$ norm, and the third for $\sigma$ is the Fubinius norm.
>
> Q3  In Theorem 2.6, should "we exist" be "there exists a set"?
> A3  yes, your phrasing is more precise.
>
> Q4 In Algorithm 1, can the authors clarify what is the main difference between X_i and $X_t$ The distinction is not clear to me.
> A4  This is a typo, X_i should be $X_{t_i}$. We have revised the corresponding paragraph.
>
> Q5 How are the integrals in Algorithm 2 computed? ...
>
> A5 Sure. First, to solve the FK equation, we need $(x_0, b, \sigma)$, which are obtained from the diffusion bridge model described earlier. Of course, these can also be obtained from any other diffusion model, as shown in the first table of the appendix. Next, we need to sample the PDE at $(x, t)$, which means we will first run the SDE. In our previous diffusion bridge model, this is based on the SDE solver, so we already have the samples $(X_t, t)$ that have been run. Then we compute the residual terms corresponding to these points, which are equations (13) and (14), and add them to the training. After training is complete, the solution $u(x,t)$ at the initial time is the mathematical expectation of the terminal distribution, which is the integral. The reason for doing this can be referenced in 3 Discussion. Overall, by doing so, we can maximize the performance of the diffusion model in estimating the mathematical expectation, without relying on the law of large numbers or ergodic theorems. For example, we do not require that $X_T$ follows a stationary distribution, and we have extended the range of $f$ in the expectation estimator.
>
> Q6 How does the computational cost of this approach compare to other MCMC approaches?
>
> A6 We believe there may be some misunderstanding regarding the main contribution of this paper. We are not making incremental improvements to diffusion models (MCMC), although we propose a sampler. What we aim to express is that the expectation estimator, i.e., (diffusion model/MCMC) + FE, will provide better estimation performance than (diffusion model/MCMC) + (LLN/ETMC). Therefore, all our usage revolves around expectation estimation, such as estimating the partition function, etc.
>
> The computational cost is divided into two parts: The first part involves solving the diffusion bridge. We use Sinkhorn's method to solve the optimal transport problem. This part is relatively fast, and you can refer to the code in our supplementary materials. Of course, if we have a pre-trained diffusion model, this computation can be completely ignored, or we can use explicit SDE samplers such as Langevin sampling. The second part, which involves training, is computationally more expensive, so we have put a lot of effort into this area. Please refer to line 275. First, we control the number of time steps for the diffusion bridge and the total number of training points. Second, we consider diagonal parametrization of $\sigma$.
>
> The last question is very valuable, and we need to provide some clarification. First, what we aim to express is that sampling is not the same as expectation estimation, because the expectation estimator is also affected by $f$ and the quality of the sampling. However, we can avoid this issue compared to traditional expectation estimators. The influence of $f$ can be illustrated using the ISING model example, where $f$ leads to a large bias due to being an exponential function. A more intuitive example is the Langevin sampling example in the appendix, where the sampler we use is biased, but our method can reduce this bias without modifying the sampler itself.
>
> This also answers your question: if we have some points sampled from the MCMC stationary distribution, we have three possible computation methods. The first is directly computing the average. The second method is to generate more points using the diffusion bridge and then compute the average. The third method is using the diffusion bridge + FK equation. We have shown that the third method is better than the first two. The main reason is that in the diffusion bridge, the parameters $b$ and $\sigma$ include all the information about the distribution. For example, in Langevin sampler, the $b$ is composed of the density($\frac12 \nabla_x \log p(x)$). Solving the PDE can better integrate this information.
>
> If you have any questions, feel free to discuss further.

---

> > ### Comment · Reviewer_xwQ7 · 2024-11-18
> >
> > A1: If the authors do not wish to indicate the definition in the main text, and want to keep it in the appendix that is fine. However, I think they should at least indicate in the main text where the definition can be found. Otherwise, how would readers know what the authors mean?
> >
> > A2: So, in the case where the metric is in the product space, we would have that $D(P\_{\theta},P\_{\eta}) = d\_x(X\_{0,\theta_3},X\_{0,\eta_3}) + d\_b(b\_{\theta_1},b\_{\eta_1}) +d\_\sigma(\sigma\_{\theta_2},\sigma\_{\eta_2})$, where the lower case ds correspond to distances in the appropriate space. Is my understanding correct?
> >
> > A5: Are you able to evaluate Equations (13) and (14) explicitly? Or are you approximating these through some partial sums? Algorithm 4 uses sums that remind me of integral approximations. If you are not using LLN or ergodic theorems for MCs to approximate such integrals, can you indicate what theoretical result confirm the validity of such evaluations and where in the text you indicate it?
> >
> > Also, how is Algorithm 4 important to your paper? I do not see it mentioned anywhere else besides the appendix?
> >
> > I do see value in the ideas posed in the paper. Furthermore, I appreciate your A6, which makes your contribution clearer. However, the presentation in the text is still not good enough to be close to acceptance. See the weaknesses I indicated above for a roadmap on how I think the presentation could be improved.
> >
> > For instance, what are you trying to show with Table 1? Can it be summarized in a manner similar to Table 5? Or rather, could you instead display it as a graphical summary?

---

> > > ### Author Response · Authors · 2024-11-18
> > > **Responses**
> > >
> > > Q1, we got it. We will pay special attention to this and mention in the main text that the definition can be found in the appendix, for example, by providing a hyperlink.
> > >
> > > Q2 yes
> > >
> > > Q3
> > > Algorithm 4 is a more precise representation of Algorithm 2 and is essentially the same algorithm. This is mentioned in line 365 of the main text, but it seems we did not provide a hyperlink. We will address this issue.
> > >
> > > In solving PDEs with PINNs, we often use the mean squared error as the loss function, which introduces integral terms. However, here we only sample within the solution domain of the PDE and use an approximation based on the LLN. Nevertheless, we did not ultimately use LLN because our expected value is derived from the initial values of the PDE solution.
> > >
> > > We have added Figure 1 to the main text, which provides a clearer explanation of our algorithm. First, we simulate the SDE, as shown on the x-t plane below. Then, we calculate the residuals at these points, corresponding to equations (13) and (14) of PINNs, and solve to obtain the PDE solution on the plane. At the very beginning of this plane lies the expected value.
> > >
> > > The efficiency of this evaluation can be verified in two ways.
> > >
> > > First, as mentioned in our paper, the integration of MCMC's key information $(X_0, b, \sigma, T)$ into the PDE solution is crucial. These four components simultaneously appear in equations (13) and (14), allowing us to provide the model with more precise information.
> > >
> > > A simple way to understand this is by considering the Langevin SDE. In this case, $b$ represents the log gradient of the density, and this information is directly integrated into equation (13) to offer a better estimate of the expectation. Similarly, $f$ is integrated into equation (14). This approach is essentially a highly integrated method combining control functions and importance sampling. The essence of control functions is to use another function $g$ to integrate information about $f$. On the other hand, importance sampling is a post-processing method that adjusts the information of $P$ through reweighting. In diffusion models, $P$ is entirely determined by $(X_0, b, \sigma, T)$.
> > >
> > > Secondly, the error of diffusion models can be analyzed in recent works. The error for using PINNs can be found in [1].
> > >
> > >
> > > However, it is worth noting that the points used in our method (on the SDE trajectory) include the entire trajectory, as shown in Figure 1. This means that even the points during the burn-in period of the MCMC algorithm can be utilized by incorporating them into equation (13).
> > >
> > > Lastly, we are addressing the challenging problem comprehensively. Including a figure might be a good approach. Thank you very much for your suggestion; we will make the corresponding revisions. The details of the Ising model experiments are provided in the Appendix 9.1. For each column in Table 1, we should provide more detailed explanations.
> > >
> > > We sincerely appreciate your valuable suggestions as we proceed with a new round of revisions. Thank you!
> > >
> > > [1]Tim De Ryck and Siddhartha Mishra. Error analysis for physics-informed neural networks (pinns) approximating kolmogorov pdes. Advances in Computational Mathematics, 48(6):79, 2022.

---

> > > > ### Comment · Reviewer_xwQ7 · 2024-11-25
> > > >
> > > > Figures 2 and 3 explain much better the previous table, and in the current state I have revised my previous rating to 3.
> > > > Are the authors using a single seed for the experiments? Usually methods are thoroughly tested, and include error quantification (e.g., standard deviations) from *multiple* runs.
> > > >
> > > > Regarding the other methods against which the authors are comparing, can they elaborate on why the labels used are MCMC-T, MCMC-R, and MCMC-C? Can the authors also indicate explicitly which methods are used for the comparison? Potentially with an accompanying reference for each.

---

> > > > > ### Author Response · Authors · 2024-11-25
> > > > > **Responses**
> > > > >
> > > > > We use fixed seeds for neural network initialization, but the sample points obtained from Gibbs sampling in MCMC do not use fixed seeds. The figures show the averaged results over multiple runs. However, it is worth noting that we use exactly the same samples for different estimators.This means that we have completely fixed the randomness in the empirical measure, ensuring fairness for all methods. We did not report the standard deviation because the table would become overly large. However, we have open-sourced our code, allowing this to be directly verified through experiments.
> > > > >
> > > > > For the current problem (calculating the partition function of a high-dimensional Ising model), there are not many expectation estimators available (despite the abundance of samplers). We chose a state-of-the-art method, the MCMC-C estimator [1], which often requires combining with the TPA method (a technique similar to importance sampling). However, due to the need for constant adjustment, it requires sampling a large number of points along the chain. When n is large, the computational complexity becomes impractical, as can be seen in cases where n>5 like https://github.com/zysophia/Doubly_Adaptive_MCMC/blob/main/data/isingcompare_complexity.csv.
> > > > >
> > > > > MCMC-R serves as a baseline because we aim to validate the matching performance of our diffusion bridge model, specifically by resampling a subset of points for averaging. This process can be seen as a reconstruction of the Markov chain (only matching points from the stationary distribution), allowing us to sample efficiently using this chain.
> > > > >
> > > > > MCMC-T directly computes the expectation by solving the FK equation using the matched diffusion bridge in MCMC-R, which is the core contribution of this paper. This method efficiently leverages the matched Markov chain and provides a superior estimation.
> > > > >
> > > > >
> > > > > [1] hahrzad Haddadan, Yue Zhuang, Cyrus Cousins, and Eli Upfal. Fast doubly-adaptive mcmc to esti- mate the gibbs partition function with weak mixing time bounds. Advances in Neural Information Processing Systems, 34:25760–25772, 2021

---

> > > > > > ### Comment · Reviewer_xwQ7 · 2024-11-26
> > > > > >
> > > > > > How many runs are the authors using? Since there are several runs, I do not understand why the standard deviations of the average MSEs are not reported in any way. At the very least they should be included as error bars, or in an appendix. This helps the reader understand how strong (if at all) the proposed method is.
> > > > > >
> > > > > > Can you also provide an answer for my other question regarding the reasoning for the labels? When I read the figures I have to keep going back to the paragraph where the methods are defined because the labels are not (to me) intuitive.
> > > > > > Is it "R" for resampling? "T" for transformed? "C" for chain? What is the logic behind the names? This is a minor point, but it makes for an unpleasant reading experience.

---

> > > > > > > ### Author Response · Authors · 2024-11-26
> > > > > > > **Responses**
> > > > > > >
> > > > > > > Ten times. Apologies, this was indeed an oversight. However, since this code was written a long time ago, we are currently re-testing, saving the relevant data, and providing a standard deviation. But, it is difficult for us to deliver a complete PDF version before the deadline. Nevertheless, our experiments are reproducible because our publicly available code can verify this. I don’t believe this is a fundamental issue.
> > > > > > >
> > > > > > > Regarding the naming issue, this is merely a notation. One explanation is MCMC-C (Correction), as this method uses different bounds to correct the estimator. MCMC-R (Resampling) refers to the estimator obtained by resampling. MCMC-T (Target) reflects our perspective of treating FKEE as a target model (DBM + FE), where the ultimate goal of this model is to estimate expectations.

---

### Meta-Review · Area_Chair_umuJ · 2024-12-19

**Metareview:**

**Summary of Discussion:**
The reviewers appreciated the novel ideas proposed in the paper, such as using the Feynman-Kac equation (FKE) and Physics-Informed Neural Networks (PINNs) to estimate mathematical expectations. However, the submission has significant issues that hinder its acceptance:

1. **Lack of Novelty and Theoretical Depth:** The reviewers noted that the paper primarily combines existing techniques (diffusion bridges, PINNs, and MCMC sampling). While innovative connections were drawn, the theoretical contributions were incremental and lacked depth. Key claims, such as reduced variance and scalability to high dimensions, were not sufficiently justified either empirically or theoretically.

2. **Insufficient Empirical Evidence:**
   - The experiments were limited to relatively low-dimensional settings (e.g., \(d=20\) or Ising model with \(15^2\) sites).
   - There was a lack of real-world applications or benchmarks against state-of-the-art methods on well-established datasets.
   - Standard deviations and error analysis were missing, leaving readers unable to assess the robustness of results.

3. **Overstated Claims:**
   - Grand claims, such as “universal samplers” and “expanding the scope of expectation estimators,” were not well-substantiated.
   - Connections to broader MCMC frameworks were vague and not explicitly supported by evidence.

4. **Presentation and Clarity Issues:**
   - Poor organization made the paper difficult to follow. Important notations and algorithms were introduced without sufficient explanation.
   - The related work section, relegated to the appendix, left readers unclear about the novelty of the contributions relative to prior work.

5. **Rebuttal Limitations:** While the authors clarified some issues during the discussion, fundamental concerns about novelty, experimental rigor, and clarity remain unresolved. Addressing these would require a substantial overhaul of the paper.

**Conclusion:**
The paper addresses an important topic but lacks the necessary clarity, rigor, and empirical support to meet the standards of ICLR. A thorough revision with a stronger theoretical foundation, robust experiments, and clearer presentation is recommended.

**Additional Comments On Reviewer Discussion:**

See above

---

### Decision · Program_Chairs · 2025-01-22

Reject